# Transfer Learning on Heterogeneous Feature Spaces for Treatment Effects Estimation

**Ioana Bica**
University of Oxford, Oxford, UK
The Alan Turing Institute, London, UK
ioana.bica@eng.ox.ac.uk

**Mihaela van der Schaar**
University of Cambridge, Cambridge, UK
University of California, Los Angeles, USA
The Alan Turing Institute, London, UK
mv472@cam.ac.uk

## Abstract

Consider the problem of improving the estimation of conditional average treatment effects (CATE) for a target domain of interest by leveraging related information from a source domain with a different feature space. This *heterogeneous transfer learning* problem for CATE estimation is ubiquitous in areas such as healthcare where we may wish to evaluate the effectiveness of a treatment for a new patient population for which different clinical covariates and limited data are available. In this paper, we address this problem by introducing several building blocks that use representation learning to handle the heterogeneous feature spaces and a flexible multi-task architecture with shared and private layers to transfer information between potential outcome functions across domains. Then, we show how these building blocks can be used to recover transfer learning equivalents of the standard CATE learners. On a new semi-synthetic data simulation benchmark for heterogeneous transfer learning we not only demonstrate performance improvements of our heterogeneous transfer causal effect learners across datasets, but also provide insights into the differences between these learners from a transfer perspective.

## 1 Introduction

Estimating the personalized effects of interventions from observational data is a fundamental problem in *causal inference* that is crucial for decision-making in many domains: in healthcare, for determining which treatments to give to patients [1], in education, for deciding which school curriculum is best for each student [2, 3], or in public policy for choosing who would benefit from job training programs [4]. Recently, a large number of machine learning methods have been proposed for estimating *conditional average treatment effects* (CATE) which enable such personalized policies [5–16].

Nevertheless, the good performance of these methods on a population of interest relies heavily on the availability of large enough observational datasets for training [17, 18]. In healthcare, for instance, this can be challenging when hospitals with few patients cannot collect enough data (e.g. a large proportion of hospitals in the USA have fewer than 100 beds, which for rare diseases can results in less than 80 training examples per year [19]). Moreover, in situations such as the COVID pandemic, each hospital will initially have very limited amount of data to learn the effectiveness of interventions from [20]. Compared to the predictive setting, this problem is exacerbated in the treatment effects setting where we need to observe both patients who are treated and not treated to be able to reliably train a model for CATE estimation to obtain personalized treatment recommendations for the intended patient population. While data from large national registries can be used to build global models for general use across hospitals, such models do not take into account the particularities of different patient populations (e.g. *different conditional outcome distributions*) and consequently can perform poorly during deployment [19, 21]. Moreover, various hospitals often record different (but overlapping) sets of patient covariates [19, 22] which makes this transfer learning problem [23]

even more challenging as we need to account for the *heterogeneous feature spaces*. Therefore, it becomes crucial to build methods for *heterogeneous transfer learning* that can leverage information from large source datasets with potentially different feature spaces to improve CATE estimation on the target datasets of interest.

We consider the Neyman-Rubin potential outcomes (PO) framework [24, 25], where for each individual we can define two *potential outcomes*, one with and one without the treatment. Out of these, we can only observe the *factual* outcome; the *counterfactual* outcome is never observed. Under the identifiability conditions of overlap and ignorability, observational data can be used to estimate the PO conditional on the patient's characteristics, which can then be used to obtain CATE. In this paper, we address the problem of heterogeneous transfer for CATE estimation, where we aim to leverage related data from a source domain with different feature space (e.g. data from another hospital) to improve CATE estimation on a target domain from which we only have few training examples.

Due to the *fundamental problem of causal inference* of not being able to observe both PO for a patient [26], heterogeneous transfer learning in the context of CATE estimation becomes significantly more challenging than for supervised learning. In addition to the feature mismatch between domains, the PO may also have different conditional distributions, as covariate relationships and their impact on patients' response to treatments cannot be expected to stay constant across hospitals/locations [19]. Moreover, as clinicians may use different criteria for assigning treatments for various patient populations, this selection bias may create discrepancies in the covariate shift induced by the treated and control populations in each domain. Consequently, we need to build an approach that can both handle the heterogeneous feature spaces, and also model the similarities and differences between both PO functions and treatment assignment mechanisms across the source and target domains.

For the binary treatment setting in a single patient population, a large number of different approaches for CATE estimation have been proposed where the main design choices involved modelling the PO functions and handling the selection bias present in observational datasets [10, 14–16, 27–29]. We discuss these in more details in Section 2. However, note that each different CATE learner has its own advantages and disadvantages in terms of the inductive biases they use for modelling the PO functions and the covariate shift induced by the selection bias, and thus, different learners will achieve better performance in various scenarios [15, 30]. Therefore, we propose a flexible approach for transfer learning, that (1) preserves the characteristics of each learner in a single domain, while (2) enabling heterogeneous feature spaces and (3) sharing information between PO functions across domains. Firstly, we introduce several building blocks that can be used to adapt the most common CATE learners [10, 15] to transfer information from a source to a target domain. These building blocks involve handling the heterogeneous feature spaces, sharing information between PO functions *across* domains and sharing information between PO functions *within* a single domain. Secondly, we show how these building blocks can be used to build heterogeneous transfer causal effect (HTCE-) learner equivalents of the most common and popular CATE learners based on neural networks [10, 15].

**Contributions.** Our contributions are three-fold (i) we define the problem of heterogeneous transfer learning in the context of CATE estimation and propose several building blocks that can be used to construct models to address this problem, (ii) we use these building blocks to construct HTCE-learner equivalents of the most common CATE learners, and (iii) we propose a new semi-synthetic data simulation and guidelines for evaluating CATE methods for heterogeneous transfer and perform extensive experiments that not only show that our HTCE-learners achieve improved performance, but also provide new insights into the differences between these learners from a transfer perspective.

## 2 Related works

We tackle the problem of heterogeneous transfer learning in the context of CATE estimation. Thus, our work straddles at the intersection of research in (1) causal inference methods for CATE estimation (2) leveraging multiple datasets for CATE estimation (3) multi-task/transfer learning and domain adaptation. Refer to Appendix A for further discussion of related works.

**CATE learners.** The estimation of CATE has received a lot of attention in the causal inference literature and several methods have been proposed to estimate the effects of binary treatments. Out of these, we consider the most popular approaches that involve using model agnostic learning strategies, also known as meta-learners, for CATE estimation [15, 31] or neural network-based models that build shared representations between the PO functions followed by outcome specific layers [10, 15, 27, 28].

The CATE meta-learners can be split into (a) one-step plug-in learners (indirect meta-learners) that estimate the PO from the observational data and then set CATE as the difference in the PO [31] and (b) two-step learners (direct meta-learners) that estimate the PO and/or the propensity score in the first step on the basis of which they build a pseudo-outcome and obtain CATE directly by regressing the input covariates on the pseudo-outcome in the second step [15, 31–33]. Refer to [15] for a more thorough classification of the different meta-learners. Alternatively, several methods based on representation learning with neural networks and multi-task learning have been proposed that involve having shared layers between the PO functions followed by outcome-specific layers. The most standard architecture for this is TARNet [10] which has been extended to allow for different types of information sharing between the PO and propensity score in [14, 15, 27]. To account for the confounding bias present in observational datasets, several approaches have been proposed to extend this model architecture by building balanced representations (treatment invariant representations) [10, 28] and/or incorporating propensity weighting to obtain unbiased estimates of the PO [29, 34]. These different approaches have their own benefits and drawback, which is why it is important to build a heterogeneous transfer learning approach that is general enough to extend all of them.

**Transfer and domain adaptation for CATE estimation.** While, the problem of transfer learning for CATE estimation has also been addressed by [35] the proposed approach considers shared feature spaces and consists of a two-stage training procedure that involves warm-start on the first domain and fine-tuning on the second domain. Alternatively, [36] proposes a CATE estimation method that can generalize to distribution shifts in the patient population in the unsupervised domain adaptation setting. However, they do not assume access to label information in the target domain and only consider a shared feature space between the two domains. In addition, [37] leverages data from multiple different environments, with shared feature spaces, to learn an invariant representation that removes the 'bad controls' which induce bias in the CATE estimation. Then, they use this invariant representation to learn shared PO functions across the different environments. Refer to Appendix A for more methods that use multiple datasets for causal inference, although for different purposes than ours.

**Multi-task/transfer learning and domain adaptation**. Methods to address these problems have been extensively studied in the predictive (supervised) setting. We describe here the works most related to ours that consider (a) shared feature space and (b) heterogeneous feature space. For shared feature spaces, methods in domain adaptation focus on handling the covariate shift, i.e. the distribution mismatch between the input features across the different domains and propose various approaches of learning domain invariant representations [38, 39] based on which they learn an outcome function shared between domains. Alternatively, multi-task/transfer learning methods propose various approaches for neural networks to learn from related tasks that involve using both shared and task (domain) specific layers [40, 41] to allow a flexible modelling of the different outcomes. To handle heterogeneous feature spaces, [22] proposes RadialGAN, a method that augments the target dataset with generated samples from the source datasets. However, RadialGAN involves training separate generators and discriminators for each domain and consequently also requires access to enough training data in the target domain. After the data generation, RadialGAN trains separate predictors in each domain that do not share information between each other. Alternatively, Wiens et al. [19] considers the problem of feature mismatch (in a specific healthcare application), but does not address the problem of distributional differences in the outcomes.

We are the first to address the problem of heterogeneous transfer for CATE estimation. We build HTCE-learners that use representation learning to handle the heterogeneous feature spaces and a multi-task architecture with shared and private layers to transfer information between PO across domains, thus also handling the case when different populations respond differently to treatments.

## 3 Problem formalism

Let random variable $X_i \in \mathcal{X}$ denote a vector of pre-treatment covariates (confounders), $W_i \in \{0, 1\}$ the assigned binary treatment and $Y_i$ a categorical or continuous observed outcome for individual $i$. Let $\pi(x) = p(W = 1 \mid X = x)$ denote the treatment assignment mechanism. As previously mentioned, we work in the Neyman-Rubin potential outcomes (PO) framework [24, 25] and we consider that each individual has two potential outcomes $Y_i(1)$ and $Y_i(0)$ for receiving and not receiving the treatment respectively. However, only one of these outcomes can be observed such that $Y_i = W_i Y_i(1) + (1 - W_i) Y_i(0)$. Let $\mu_1(x) = \mathbb{E}[Y(1) \mid X = x]$ and $\mu_0(x) = \mathbb{E}[Y(0) \mid X = x]$ be the PO functions.

Our aim is to estimate the conditional average treatment effect (CATE):
$$\tau(x) = \mathbb{E}[Y(1) - Y(0) \mid X = x] = \mu_1(x) - \mu_0(x) \qquad (1)$$
which is the difference between expected outcomes for an individual with covariates $X = x$. Let $\eta = (\mu_0(x), \mu_1(x), \pi(x))$ be the nuisance functions for this CATE estimation problem.

Assume access to a source dataset $\mathcal{D}^R = \{(X_i^R, W_i, Y_i)\}_{i=1}^{N_R}$ and a target dataset $\mathcal{D}^T = \{(X_i^T, W_i, Y_i)\}_{i=1}^{N_T}$. Different domains in applications such as healthcare, have heterogeneous feature spaces such that $X_i^R \in \mathbb{R}^{D_R}$ and $X_i^T \in \mathbb{R}^{D_T}$, where $D_R \neq D_T$ are the dimensions of the feature spaces. We also consider that the source and target domains have different distributions $p(X^R) \neq p(X^T)$ (due to their heterogeneous feature spaces), different treatment assignment mechanisms $p(W = 1 \mid X^R) \neq p(W = 1 \mid X^T)$ and different conditional distributions for PO, $p(Y(w) \mid X^R) \neq p(Y(w) \mid X^T)$. This results in different joint distributions $p(X^R, W, Y) \neq p(X^T, W, Y)$ which is representative of hospitals recording different types of patient data where the relationships between patient covariates, treatments and outcomes can change across diseases and locations [19, 42]. Nevertheless, we implicitly assume that there is a shared structure between these conditional distributions across domains to enable transfer.

Our aim is to estimate conditional average treatment effects (CATE) for the target domain:
$$\tau^T(x) = \mu_1^T(x^T) - \mu_0^T(x^T), \qquad (2)$$
by using both the source $\mathcal{D}^R$ and target dataset $\mathcal{D}^T$. In particular, we want to improve the estimation in the target domain by leveraging information from the source domain. This is useful in the setting where the target dataset is much smaller than the source one $N_T << N_R$ and we can leverage shared structure between the source and target outcome response functions: $\mu_0^R(x^R), \mu_0^T(x^T)$ and $\mu_1^R(x^R), \mu_1^T(x^T)$ and treatment assignment mechanisms $\pi^R(x^R), \pi^T(x^T)$. To be able to identify the causal effects from observational data, we make the standard assumptions for both domains.

**Assumption 1.** *(Unconfoundedness) There are no unobserved confounders, such that the treatment assignment and PO are conditionally independent given the covariates:* $Y(0), Y(1) \perp\!\!\!\perp W \mid X^T$ *and* $Y(0), Y(1) \perp\!\!\!\perp W \mid X^R$.

**Assumption 2.** *(Overlap)* $\pi^T(x^T) = p(W = 1 \mid X^T = x^T) > 0, \forall x^T \in \mathcal{X}^T$ *and* $\pi^R(x^R) = p(W = 1 \mid X^R = x^R) > 0, \forall x^R \in \mathcal{X}^R$.

# 4 Building blocks for CATE transfer learners

In this section, we propose building blocks that enable a flexible transfer approach for CATE learners. The challenge in this setting is threefold as we need to (1) handle heterogeneous feature spaces between the source and target domains (2) share information between PO functions across source and target datasets $(\mu_1^R, \mu_1^T)$ and $(\mu_0^R, \mu_0^T)$ as well as (3) share information between PO functions within a single domain $(\mu_0^R, \mu_1^R)$ and $(\mu_0^T, \mu_1^T)$.

We start by addressing (1) and (2) and show how the proposed building blocks can be used to obtain transfer approaches for the most common meta-learning strategies in the treatment effects literature [15]. Then, we propose a building block for addressing (3) to obtain transfer CATE learners that use shared layers and outcome specific layers for the potential outcome functions in each domain [10].

## 4.1 Handling heterogeneous feature spaces between source and target domains

Consider the following split for the source and target covariates $X^R = (X^s, X^{p_R})$ and $X^T = (X^s, X^{p_T})$ such that we have a set of features private (specific) to the source dataset $X^{p_R} \in \mathbb{R}^{D_{p_R}}$, a set of features private to the target dataset $X^{p_T} \in \mathbb{R}^{D_{p_T}}$ and a set of shared features between the two datasets $X^s \in \mathbb{R}^{D_S}$. To handle the heterogeneous features spaces between the source and target datasets we propose using several encoders to create a common representation that can be used as input to the different transfer CATE learners.

Let $\phi^{p_R}(x^R) : \mathbb{R}^{D_R} \to \mathbb{R}^{D_p}$ and $\phi^{p_T}(x^T) : \mathbb{R}^{D_T} \to \mathbb{R}^{D_p}$ be domain-specific (private) encoders that map the heterogeneous input features to a representation of size $D_p$, such that $\phi^{p_T}(x^T) = z^{p_T}$ and $\phi^{p_R}(x^R) = z^{p_R}$. Moreover, let $\phi^s(x^s) : \mathbb{R}^{D_S} \to \mathbb{R}^{D_s}$ be a shared encoder that maps the shared features between the source and target domains into a representation of size $D_s$ such that $\phi^s(x^s) = z^s$.

As illustrated in Figure 1, a source example $x^R$ is encoded to $[z^s||z^{p_R}]$ and a target example $x^T$ to $[z^s||z^{p_T}]$, where $||$ denotes concatenation and where both representations have size $D_s + D_p$. Note that an alternative approach would have been to use the domain-specific encoders $\phi^p$ only for the private features $x^{p_R}$ and $x^{p_T}$. However, inputting the shared features through both types of encoders allows us to learn relationships between them that are shared across the different domain, as well as interactions which are domain-specific.

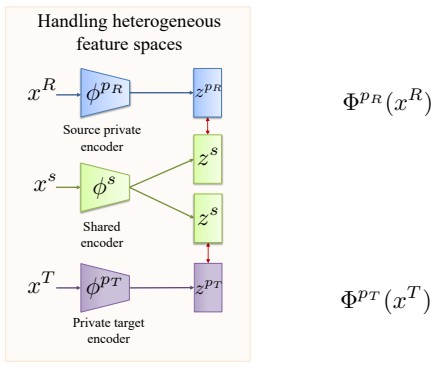

$\Phi^{p_R}(x^R)$

$\Phi^{p_T}(x^T)$

To discourage redundancy and ensure that $z^p$ and $z^s$ encode different information from the input features, we propose using a regularization loss that enforces their orthogonality [39]:

$$\mathcal{L}_{\text{orth}_z} = \|\zeta^{s\top}\zeta^{p_R}\|_F^2 + \|\zeta^{s\top}\zeta^{p_T}\|_F^2 \qquad (3)$$

where $\zeta^{p_R}, \zeta^{p_T}$ and $\zeta^s$ are matrices whose rows are the private $z^{p_R}$, $z^{p_T}$ and shared $z^s$ representations for the source and target examples respectively, and $\|\cdot\|_F^2$ is the squared Frobenius norm.

Figure 1: Building block for handling the heterogeneous feature space of the source and target domains.

### 4.2 Sharing information between potential outcomes response functions across domains

As treatment responses can vary between different patient populations, it is important to build a transfer approach that enables learning target-specific outcome functions, while also sharing information from the source domain. We propose a building block for sharing information between PO functions across domains that is inspired by the FlexTENet architecture [14] and by works in multitask learning [41] and that involves having private layers (subspaces) for each domain as well as shared layers.

As shown in Figure 2, for each treatment $w \in \{0, 1\}$, we consider a model architecture for estimating its PO functions in the source and target domains $\mu_w^R$ and $\mu_w^T$ that consists of $L$ layers, each having a shared and two private subspaces (one for each domain). For simplicity, we consider the same number of hidden dimensions for each shared and private subspace. Let $\tilde{h}_{w,l}^{p_R}$, $\tilde{h}_{w,l}^{p_T}, \tilde{h}_{w,l}^s$ be the inputs and $h_{w,l}^{p_R}, h_{w,l}^{p_T}, h_{w,l}^s$ the outputs of the $l^{th}$ layer. For $l > 1$, similarly to [14], the inputs to the $(l+1)^{th}$ layer are obtained as follows: $\tilde{h}_{w,l+1}^{p_R} = [h_{w,l}^s||h_{w,l}^{p_R}], \tilde{h}_{w,l+1}^{p_T} = [h_{w,l}^s||h_{w,l}^{p_T}], \tilde{h}_{w,l+1}^s = [h_{w,l}^s]$. For $l = 1$, we set $\tilde{h}_{w,1}^{p_R} = \Phi^R(x^R), \tilde{h}_{w,1}^{p_T} = \Phi^T(x^T)$, and $\tilde{h}_{w,1}^s = \tilde{h}_{w,1}^{p_R}$ when using an example from the source domain or $\tilde{h}_{w,1}^s = \tilde{h}_{w,1}^{p_T}$ when using an

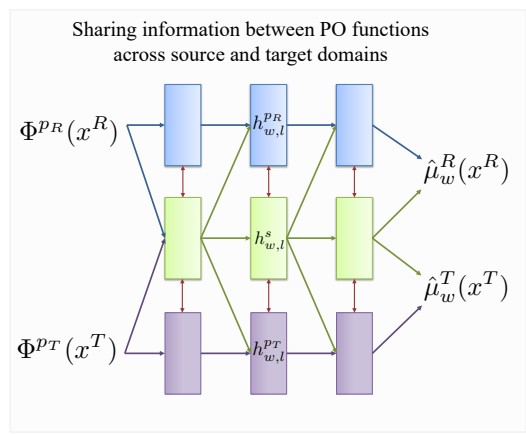

$\Phi^{p_R}(x^R)$    $h_{w,l}^{p_R}$    $\hat{\mu}_w^R(x^R)$

$h_{w,l}^s$

$\Phi^{p_T}(x^T)$    $h_{w,l}^{p_T}$    $\hat{\mu}_w^T(x^T)$

Figure 2: Building block for sharing information between PO across domains.

example from the target domain, where $\Phi^R(\cdot)$ and $\Phi^T(\cdot)$ are input representations. When sharing the encoders from Section 4.1 for both treatments, we set $\Phi^R(x^R) = [z^s||z^{p_R}]$ and $\Phi^T(x^T) = [z^s||z^{p_T}]$. However, as we will see in Section 5.1, this input representation is CATE learner specific and can be extended (see Section 5.2) by adding more representation layers to share information between PO functions within each domain. For the last layer $L$, we build $h_{w,L}^s, h_{w,L}^{p_R}, h_{w,L}^{p_T}$ to each have the same dimension as the potential outcome $y$.

Overall, let $g_w^R, g_w^T$ be the hypothesis functions estimating the potential outcomes in the source and target domains respectively, such that $g_w^R(\Phi^R(x^R)) = \psi(h_{w,L}^{p_R} + h_{w,L}^s)$ and $g_w^T(\Phi^T(x^T)) = \psi(h_{w,L}^{p_T} + h_{w,L}^s)$, where $\psi$ is the linear function for continuous outcomes and sigmoid function for binary ones. This allows us to define the following loss function for estimating the PO:

$$\mathcal{L}_y = \sum_{i=1}^{N_R} l(y_i, g_{w_i}^R(\Phi_{w_i}^R(x_i^R))) + \sum_{i=1}^{N_T} l(y_i, g_{w_i}^T(\Phi_{w_i}^T(x_i^T))) \qquad (4)$$

where $l(\cdot, \cdot)$ can be the mean squared error for continuous outcomes and the cross-entropy for binary outcomes. Moreover, to discourage redundancy between the shared and private layers, we apply an orthogonal regularization loss, similar to [14, 41]. Let $m^s_{w,l-1}, m^{p_R}_{w,l-1}, m^{p_T}_{w,l-1}$ be the dimensions of $h^s_{w,l-1}, h^{p_R}_{w,l-1}, h^{p_T}_{w,l-1}$ respectively, i.e. the outputs of the $(l-1)^{th}$ layer. Let the weights in the $l^{th}$ layer be $\theta^s_{w,l} \in \mathbb{R}^{m^s_{w,l-1} \times m^s_{w,l}}$, $\theta^{p_R}_{w,l} \in \mathbb{R}^{(m^s_{w,l-1} + m^{p_R}_{w,l-1}) \times m^s_{w,l}}$ and $\theta^{p_T}_{w,l} \in \mathbb{R}^{(m^s_{w,l-1} + m^{p_T}_{w,l-1}) \times m^s_{w,l}}$. This allows us to use the following orthogonal regularizer:

$$\mathcal{L}_{\text{orth}_{PO}} = \sum_{w \in \{0,1\}} \sum_{l=1}^{L} \|\theta^s_{w,l}{}^\top \theta^{p_R}_{w,l,1:m^s_{l-1}}\|_F^2 + \|\theta^s_{w,l}{}^\top \theta^{p_T}_{w,l,1:m^s_{w,l-1}}\|_F^2 \tag{5}$$

A similar approach can be used to share information between the propensity estimation functions $(\hat{\pi}(x))$ that are used by some of the meta-learners. See appendix B for details.

## 5 Heterogeneous transfer causal effect learners

Using these building blocks of handling the heterogeneous feature spaces and sharing information about PO functions across domains, we now propose a transfer learning alternative for the standard meta-learners and neural networks (NNs) based CATE estimators, which we refer to as Heterogeneous Transfer Causal Effect (HTCE) learners. See Appendix C for the pseudo-code for the HTCE-learners.

### 5.1 Heterogeneous transfer learning for CATE meta-learners

Consider NN-based implementations for the nuisance functions $\eta = (\mu_0(x), \mu_1(x), \pi(x))$ of each CATE meta-learner. Based on the taxonomy of meta-learners described in [15] we divide them into *one-step plug-in learners* and *two-step learners* and provide HTCE- equivalents for both.

**One-step plug-in learners.** One-step plug-in learners estimate $\hat{\mu}_1$ and $\hat{\mu}_0$ and then compute CATE as $\hat{\tau}(x) = \hat{\mu}_1(x) - \hat{\mu}_0(x)$. The most common strategies are the T-learner which does not share information between PO functions and the S-learner which does [31]. Here, we consider the simplest S-learner that uses the treatment as an additional feature, while in Section 5.2 we explore more complex strategies for sharing information between PO functions within a single domain.

For our *HTCE-T-learner* (Figure 3 (a)), we build a model where there is no parameter sharing between $(g^R_0, g^R_1)$ and between $(g^T_0, g^T_1)$. We consider treatment-specific encoders $\phi^{p_R}_w, \phi^{p_T}_w, \phi^s_w$ for the heterogeneous feature spaces, such that $\Phi^R_w(x^R) = [\phi^s_w(x^s)||\phi^{p_R}_w(x^R)]$ and $\Phi^T_w(x^T) = [\phi^s_w(x^s)||\phi^{p_T}_w(x^T)]$. To enable transfer, we use the approach described in Section 4.2 to share information between the hypothesis functions $(g^R_0(\Phi^R_0(x^R)), g^T_0(\Phi^T_0(x^T)))$ and $(g^R_1(\Phi^R_1(x^R)), g^T_1(\Phi^T_1(x^T)))$. Alternatively, for the *HTCE-S-learner* (Figure 3 (b)), we propose concatenating the treatment $w$ to the shared input features $x^s$ and using shared layers to obtain the PO functions in each domain. Thus, we build input representations $\Phi^R(x^R, w) = [\phi^s(x^s, w)||\phi^{p_R}(x^R)]$, $\Phi^T(x^T, w) = [\phi^s(x^s, w)||\phi^{p_T}(x^T)]$ and we enable transfer by having private and shared layers between the hypothesis functions $g^R(\Phi^R(x^R, w))$ and $g^T(\Phi^T(x^T, w))$ that share parameters for both treatments within a single domain.

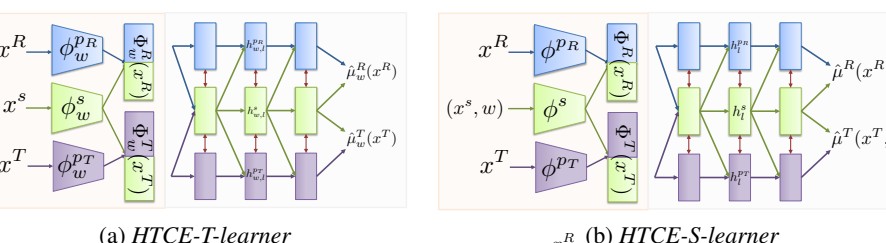

(a) *HTCE-T-learner*          (b) *HTCE-S-learner*

Figure 3: HTCE-one-step plug-in learners. Notice that the HTCE-T-learner uses treatment specific feature encoders and hypothesis functions in each domain, while the HTCE-S-learner concatenates the treatment to the shared features and shares its components for both treatments in each domain.

**Two-step learners.** The two-step (direct) learners consist of a first stage that involves obtaining plug-in estimates $\hat{\eta}$ of the nuisance parameters $\eta = (\mu_0, \mu_1, \pi)$ which are used to compute the pseudo-outcome $\tilde{Y}_{\hat{\eta}}$ and a second stage that involves regressing $\tilde{Y}_{\hat{\eta}}$ on X directly to obtain $\hat{\tau}$. Several

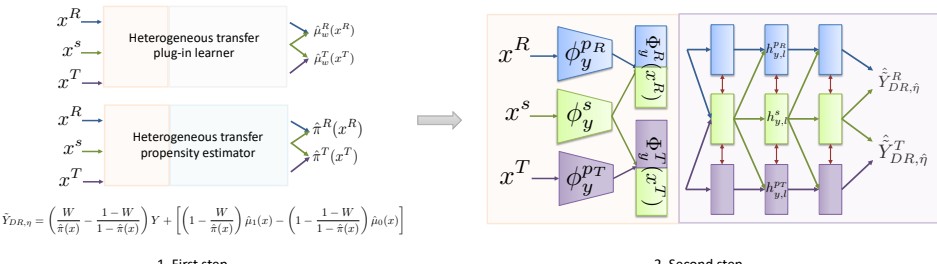

1. First step          2. Second step

Figure 4: *HTCE-DR-learner*. The first step obtains estimates of $\hat{\eta}$ using our hetereogeneous transfer learning approach that shares information between $(\hat{\mu}_w^R, \hat{\mu}_w^T)$ and $(\hat{\pi}^R, \hat{\pi}^T)$, which are then used to compute the pseudo-outcomes $\tilde{Y}_{DR,\eta}$. The second stage regresses the input on the pseud-outcomes and builds a hetereogeneous transfer approach to share information between $(\hat{\tilde{Y}}_{DR,\eta}^R, \hat{\tilde{Y}}_{DR,\eta}^T)$.

strategies based on regression adjustment (RA), propensity weighting (PW) or doubly robust (DR) meta-learner [15] have been proposed to compute pseudo-outcomes $\tilde{Y}_{\hat{\eta}}$ which result in unbiased estimates of CATE when $\eta$ is known: $\tau(x) = \mathbb{E}[\tilde{Y}_{\hat{\eta}} \mid X = x]$. These meta-learners have different theoretical properties depending on the sample size and the selection bias present in observational datasets [15], which is why it is important to build a transfer approach to extend all of them.

We describe here how to build the *HTCE-DR-learner* (Figure 4) and note that a similar approach can be used for to obtain the *HTCE-RA-learner*, and the *HTCE-PW-learner* which use a subset of nuissance parameters to compute the pseudo-outcomes. For the first stage of the *HTCE-DR-learner* of estimating the nuisance parameters $\hat{\eta}$, we use an approach similar to the HTCE-T-learner where each nuisance function has its own parameters. To handle the heterogeneous feature spaces, we consider treatment-specific feature encoders for the potential outcome functions $\phi_w^{p_R}, \phi_w^{p_T}, \phi_w^s$ and additional feature encoders for the propensity estimation $\phi_\pi^{p_R}, \phi_\pi^{p_T}, \phi_\pi^s$ such that $\Phi_w^R(x^R) = [\phi_w^s(x^s)||\phi_w^{p_R}(x^R)]$, $\Phi_w^T(x^T) = [\phi_w^s(x^s)||\phi_w^{p_T}(x^T)]$ and $\Phi_\pi^R(x^R) = [\phi_\pi^s(x^s)||\phi_\pi^{p_R}(x^R)]$, $\Phi_\pi^T(x^T) = [\phi_\pi^s(x^s)||\phi_\pi^{p_T}(x^T)]$. Using the approach described in Section 4.2 and Appendix B, we share information across domains between each nuisance function: $(g_w^R(\Phi_w^R(x^R)), g_w^T(\Phi_w^T(x^T)))$ and $(g_\pi^R(\Phi_\pi^R(x^R)), g_\pi^T(\Phi_\pi^T(x^T)))$ to estimate $\hat{\eta}$. We then compute the pseudo-outcomes $\tilde{Y}_{DR,\hat{\eta}}$ as follows [15]:

$$\tilde{Y}_{DR,\eta} = \left( \frac{W}{\hat{\pi}(x)} - \frac{1-W}{1-\hat{\pi}(x)} \right) Y + \left[ \left( 1 - \frac{W}{\hat{\pi}(x)} \right) \hat{\mu}_1(x) - \left( 1 - \frac{1-W}{1-\hat{\pi}(x)} \right) \hat{\mu}_0(x) \right] \quad (6)$$

For the second stage of regressing the input on the pseudo-outcomes $\tilde{Y}_{DR,\hat{\eta}}^R, \tilde{Y}_{DR,\hat{\eta}}^T$ directly, we build a similar transfer architecture with feature encoders $\phi_y^{p_R}, \phi_y^{p_T}, \phi_y^s$ to handle the heterogeneous feature, followed by hypothesis functions with private and shared layers $(g_y^R(\Phi_y^R(x^R)), g_y^T(\Phi_y^T(x^T)))$, where $\Phi_y^R(x^R) = [\phi_y^s(x^s)||\phi_y^{p_R}(x^R)]$, $\Phi_y^T(x^T) = [\phi_y^s(x^s)||\phi_y^{p_T}(x^T)]$.

## 5.2 Sharing information between potential outcome functions within each domain

In previous sections, we focused on sharing information between PO functions *across* domains $(\mu_1^R, \mu_1^T)$ and $(\mu_0^R, \mu_0^T)$. However, many methods in the causal inference literature for estimating treatment effects propose different strategies for also sharing information between PO functions *within* a single domain $(\mu_0^R, \mu_1^R)$ and $(\mu_0^T, \mu_1^T)$ [10, 15, 27]. The most common one is the TARNet architecture [10] which uses several layers to build a shared representation, followed by outcome-specific layers.

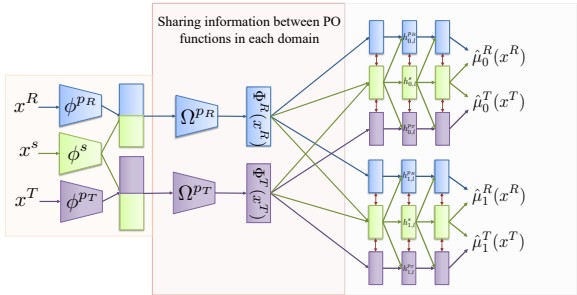

Figure 5: *HTCE-TARNet*.

To obtain *HTCE-TARNet* (Figure 5), we propose using feature encoders $\phi^{p_R}, \phi^{p_T}, \phi^s$ that are shared across the different treatments in each

domain. Then, we construct another building block that involves adding several representation layers on top of the feature embeddings in each domain using domain-specific encoders $\Omega^{p_T}$ and $\Omega^{p_R}$. These representation layers are domain specific as their purpose is to share information between the PO functions within a single domain. Thus, we obtain $\Phi^R(x^R) = \Omega^{p_R}([\phi^s(x^s)||\phi^{p_R}(x^R)])$, $\Phi^T(x^T) = \Omega^{p_T}([\phi^s(x^s)||\phi^{p_T}(x^T)])$, which are used as input to the corresponding (treatment-specific) hypothesis functions $g_w^R(\Phi^R(x^R))$ and $g_w^T(\Phi^T(x^T))$.

This approach can be easily extended to recover HTCE- equivalents of other SNet architecture [15] that build additional representation layers to share information between PO (and propensity) estimation within each single domain [27]. Moreover, several methods have been proposed to account for the selection bias in the observational dataset by building balancing representations [10, 27, 28]. This can also be achieved for our HTCE-learners by enforcing regularization on top of the learnt domain-specific representations: $\Omega^{p_R}(\cdot)$ and $\Omega^{p_T}(\cdot)$. Overall, we have built the HTCE-learners to preserve the characteristics of the different CATE-learners in a single domain, but to handle the heterogeneous feature spaces and to share information between outcome functions across domains.

## 6 Evaluation

### 6.1 Experimental set-up

**Semi-synthetic data generation.** Given that counterfactuals, and subsequently ground truth causal effects are never observed in real-datasets, semi-synthetic data is needed to be able to evaluate methods for estimating CATE. While several benchmarks have been proposed for the standard CATE estimation problem [15, 17] in a single domain, none exist for the heterogeneous transfer learning setting. Thus, we propose a new semi-synthetic data simulation to evaluate our HTCE-learners against the benchmarks. Let $X^R = (X^s, X^{p_R})$ and $X^T = (X^s, X^{p_T})$ be patient features from real source and target datasets respectively. We simulate outcomes as follows:

$$Y^R(w) = \alpha \sum_{j=1}^{D_S}(v_{w,j}^s X_j^s)/D_S + (1-\alpha)\left[\beta \sum_{j=1}^{D_{p_R}}(v_{w,j}^{p_R} X_j^{p_R})/D_{p_R} + (1-\beta)\sum_{j=1}^{D_R}(v_j^R X_j^R)/D_R\right] + \epsilon_R$$
(7)

$$Y^T(w) = \alpha \sum_{j=1}^{D_S}(v_{w,j}^s X_j^s)/D_S + (1-\alpha)\left[\beta \sum_{j=1}^{D_{p_T}}(v_{w,j}^{p_T} X_j^{p_T})/D_{p_T} + (1-\beta)\sum_{j=1}^{D_T}(v_j^T X_j^T)/D_T\right] + \epsilon_T$$
(8)

where $v_{w,j}^s, v_{w,j}^{p_R}, v_{w,j}^{p_T}, v_j^R, v_j^T \sim \mathcal{N}(-10,10)$, $\epsilon_R, \epsilon_T \sim \mathcal{N}(0,0.1)$ and the parameters $\alpha$ and $\beta$ control the amount of sharing between PO across and within domains respectively. For each domain, we simulate the treatment assignment using $W \mid X \sim \text{Bernoulli}(\text{Sigmoid}(\kappa(Y(1) - Y(0))))$, where for $\kappa = 0$ there is no selection bias and the treatments are assigned randomly, while a high $\kappa$ indicates high selection bias. Unless otherwise specified, we set $\alpha = 0.5, \beta = 0.5$ and $\kappa_R = 1, \kappa_T = 1$.

**Datasets.** We perform experiments using patient features from three real datasets with diverse characteristics (e.g. number of features, proportion of categorical features): Twins [43], TCGA [44] and MAGGIC [45]. From the real patient features, the treatment assignments and corresponding outcomes are simulated as described. Refer to Appendix D for details of the datasets and of the way the source and target domain and the shared and private features are obtained. Due to space limitation, we report here the results on Twins. See Appendix E for the results on the other datasets.

**CATE learners and benchmark methods.** The following CATE learners are used in our experiments: S-Learner, T-Learner, DR-Learner and TARNet. We consider the following benchmarks: (1) training the CATE learners only on the target dataset, (2) using only the shared features between the source and target datasets and (3) using RadialGAN [22] to 'translate' the source dataset into the target domain. We compare these against our HTCE-learners that leverage the full source and target datasets for training. To ensure a fair comparison, we fixed equivalent hyperparameters in terms of number of layers and units in each layer for the CATE-learners trained on data from the different benchmarks and our HTCE-learners. We describe the hyperarameter used and provide implementation details in Appendix D. We evaluate the different methods using the the Root Mean Squared Error of estimating $\tau^T(x^T)$, also known as the precision of estimating heterogeneous effects (PEHE) [5]. The results are averaged across 10 runs for which we report the mean PEHE and its standard error.

## 6.2 Results and discussion

**Benchmarks comparison and source of gain**. We first evaluate the HTCE-learners[1] against the benchmarks and perform a source of gain analysis to evaluate the importance of our different components and loss functions. We evaluate the impact of removing the shared and private layers that enable sharing information between the PO functions across

Table 1: Benchmarks comparison and source of gain analysis in terms of PEHE on Twins dataset.

| Learner | S-Learner | T-Learner | DR-Learner | TARNet |
|---|---|---|---|---|
| Target | $0.30 \pm 0.03$ | $0.23 \pm 0.02$ | $0.20 \pm 0.03$ | $0.19 \pm 0.02$ |
| Shared features | $0.46 \pm 0.10$ | $0.47 \pm 0.10$ | $0.46 \pm 0.09$ | $0.46 \pm 0.09$ |
| RadialGAN | $0.21 \pm 0.02$ | $0.2 \pm 0.02$ | $0.17 \pm 0.01$ | $0.19 \pm 0.02$ |
| HTCE - PO sharing | $0.18 \pm 0.03$ | $0.15 \pm 0.01$ | $0.11 \pm 0.01$ | $0.12 \pm 0.01$ |
| HTCE - $\mathcal{L}_{\text{orth}_z}$ | $0.14 \pm 0.01$ | $0.08 \pm 0.01$ | $0.07 \pm 0.01$ | $0.11 \pm 0.01$ |
| HTCE - $\mathcal{L}_{\text{ortho}_{PO}}$ | $0.15 \pm 0.01$ | $0.11 \pm 0.01$ | $0.10 \pm 0.01$ | $0.11 \pm 0.01$ |
| HTCE (ours) | $\mathbf{0.12} \pm 0.01$ | $\mathbf{0.06} \pm 0.01$ | $\mathbf{0.05} \pm 0.01$ | $\mathbf{0.09} \pm 0.01$ |

domains (HTCE - PO sharing), removing the orthogonal regularization loss for the learnt shared and private feature representations in Equation 3 (HTCE - $\mathcal{L}_{\text{orth}_z}$) and removing orthogonal regularization loss from the PO layers in Equation 5 (HTCE - $\mathcal{L}_{\text{ortho}_{PO}}$). Table 1 reports results on the Twins dataset. Note that our HTCE-learners achieve better performance compared to the baselines and that each component we propose brings performance improvements. When analyzing the performance across the different CATE learners we notice that the S-Learner achieves worse performance due to its strong inducting bias of fully-sharing information between PO functions within each domains, which becomes even more restrictive in the transfer setting.

**Varying the information sharing between domains**. To further gain insights into the differences between the benchmarks under different settings, we vary the parameter $\alpha$, which controls the amount of information shared between the PO in the source and target dataset. Figure 6 (top) reports the results for this analysis on the Twins dataset. Note that our flexible approach for information sharing between PO functions across domains achieves good performance both when the PO functions are significantly different between the source and target dataset ($\alpha = 0.1$) and also when they have the same functional form and only depend on the shared features ($\alpha = 1.0$). As a sanity check, we also notice that for ($\alpha = 1.0$) the performance of the HTCE-learners matches the one of the corresponding CATE learners trained only on the shared features between the two domains. Moreover, note that for this setting of $\alpha = 1.0$ when the PO only depend on the shared features between the two domains, the different CATE learners trained only on the target dataset have a decrease in performance as they also need to perform the task of implicit feature selection with limited amount of data.

**Varying the target dataset size.** We also evaluate performance when varying the size of the target dataset $N_T$. We report in Figure 6 (bottom) the results on the Twins datasets. Note that while our HTCE-learners still bring benefits for all values of $N_T$, the most performance improvements are when the target dataset size is small and we get diminishing returns as $N_T$ increases. Moreover, note that TARNet trained only on the target dataset reaches the performance of the HTCE-TARNet learner with less amount of data compered to T-learner trained on the target dataset vs. the HTCE-T-learner.

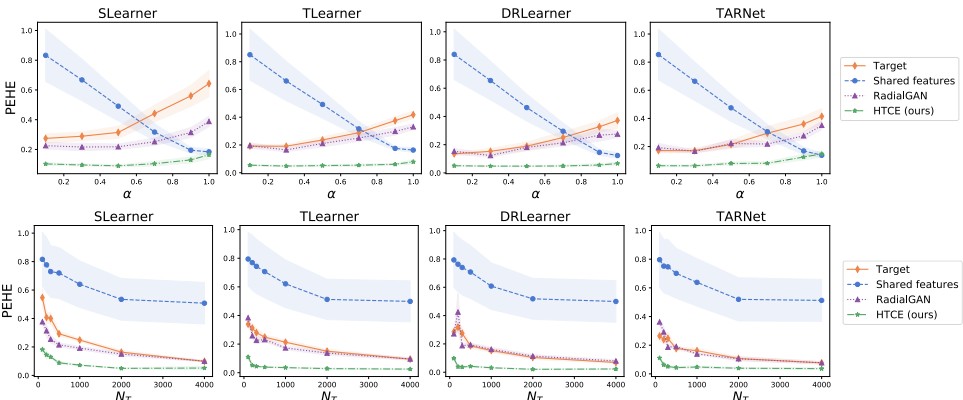

Figure 6: Performance comparison on Twins when varying the information sharing between PO functions across domains through $\alpha$ (top) and the size of the target dataset through $N_T$ (bottom).

---

[1]The code for the HTCE-learners can be found at https://github.com/vanderschaarlab and at https://github.com/ioanabica/HTCE-learners

**Effect of selection bias.** Finally, we investigate how the selection bias in the source and target datasets impact transfer performance. We consider three setting for the selection bias in the source dataset $\kappa_R = 0.0$ for random treatment assignment, $\kappa_R = 2.0$ for moderate and $\kappa_R = 10.0$ for strong selection bias. For each setting of $\kappa_R$, we vary the selection bias in the target dataset $\kappa_T$ from $\kappa_T = 0.0$ to $\kappa_T = 10.0$ and report the results on Twins for the DR-learner in Figure 7. See Appendix E for results on the

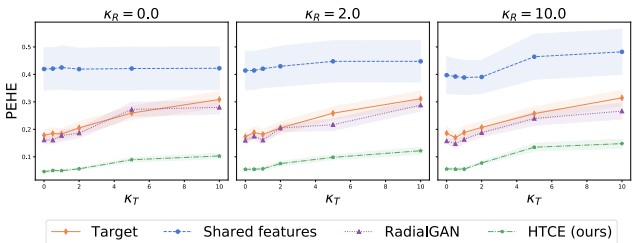

Figure 7: Performance comparison on Twins for DR-learner when varying the selection bias $\kappa_R$ and $\kappa_T$ in the source and target datasets.

other learners and datasets. Note that our HTCE-DR-learner has consistent performance when increasing the selection bias in both datasets and when there are significant discrepancies in the treatment assignment mechanism between the source and target datasets.

## 7 Discussion

We proposed heterogeneous transfer causal effect (HTCE) learners capable of improving the estimation of CATE on a target domain by leveraging data from a related source domain. This represents the first work that addresses the heterogeneous transfer learning problem for CATE estimation which is prevalent in many causal inference application areas where the personalized effects of interventions for a new population need to be estimated from small target datasets with different features than the available source datasets [3, 19]. We hope that the insights gained from this paper, in terms of the need for sharing information between PO functions to enable transfer and the differences between CATE learners in the transfer setting, will help guide further model development for this problem.

In terms of limitations and directions for future work, while we focused on the binary treatment setting which is the most common in the causal inference literature, addressing the problem of heterogeneous transfer learning is equally important for more complex treatment settings such as treatments with dosage [46, 47], treatment combinations [48, 49] and treatments in the temporal setting [50-54]. Moreover, while we demonstrate the benefits of our HTCE-learners experimentally, future work should also provide theoretical guarantees on their performance in terms of the similarities between the source and target datasets. We provide further discussion of possible extensions and limitations of our method in Appendix F. Finally, we acknowledge that in areas such as healthcare, acting based on incorrect estimates from such causal inference models can have life-threatening implications. Because of this, it becomes crucial that such models undergo extensive testing through clinical trials and expert validation and that they are only used for decision support alongside clinicians.

## Acknowledgments and Disclosure of Funding

We would like to thank the reviewers for their valuable feedback. Moreover, we would also like to thank Alicia Curth for insightful discussions. The research presented in this paper was supported by The Alan Turing Institute, under the EPSRC grant EP/N510129/1, by the US Office of Naval Research (ONR), and by the National Science Foundation (NSF) under grant number 1722516.

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
