# A  Expanded related works

In this section, we further expand on the related works described in Section 2.

**Additional methods that leverage multiple datasets for causal inference.** While several other methods propose leveraging multiple datasets to improve CATE estimation, these focus on combining observational data with randomized data from clinical trials with the aim of handling the bias from hidden confounders [55–57]. Moreover, all of these methods learn potential outcome functions that are shared between the different domains. Conversely, we assume that there are no hidden confounders and propose an approach that can share information from a source domain to improve CATE estimation on a target domain of interest.

A different strand of work aims to address the problem of identifiability of causal effects for an observational dataset of interest by leveraging data from other datasets. In particular, [58] investigates how to transfer average treatment (causal) effects obtained from experimental data (such as randomized clinical trials) to an observed population that may have different distribution of covariates, treatments and outcomes and where the causal relationship of interest cannot be identified using only the observational data. The paper assesses under which conditions such average causal effects can be transported according to the differences between the randomized and observational data. The authors also provide a brief discussion of how to transfer these average causal estimates between observational datasets. Alternatively, [59] aims to help identify the average effects of interventions on a target population of interest by integrating multiple types of auxiliary data: data from a randomized study on the same population, data from an observational study on the same population, selection biased data from the same population and data from a randomized study from a different population. All of the auxiliary datasets have the same set of features as the target dataset. Both [58, 59] consider this transportability problem of average treatment effects in the context of causal diagrams. This is different from our set-up for CATE estimation where we assume that the potential outcomes are identifiable in both the source and target domains. Moreover, we also handle the case of heterogeneous feature spaces and only assume access to a source domain larger than a target domain, without putting any restrictions on whether this data is experimental or observational.

**Transfer learning and domains adaptation in the predictive setting**. Another method loosely related to ours is the one of [60], which considers the case of having both heterogeneous feature and label spaces in the context of natural language processing and proposes a method that learns a common embedding for the features and labels and then builds a mapping between them. In this paper, we consider a shared label space. In addition, [61] uses a causal approach to address the problem of domain adaptation for the predictive setting, where it considers labelled data in one or more source domains, unlabelled data in the target domain, same feature spaces between the source and target domains and aims to learn predictive functions that are invariant to the changes between domains to be able to reliably estimate the outcomes in the target domain. [61] proposes tackling this unsupervised domain adaptation by modelling the different distributions between the source and target domains as different contexts of a single underlying system (where the context represents interventions causing the distribution shifts between domains, such that the source and target domains come from different interventions).

Table 2 provides a comparison of our method with the most relevant related works. Note that our heterogeneous transfer learning approach is designed to share information between one source domain and a target domains. Moreover, we assume access to labels on the target domain and build target-specific outcome functions that share a common structure with the source domain.

| | Method | Number of domains | Heterogeneous feature spaces | Heterogeneous label spaces | Access to labels on target domain | Domain-specific outcome functions |
|---|---|---|---|---|---|---|
| Causal Inference | CATE learners [10, 14, 15] | 1 | No | No | No | No |
| | Johansson et al. [36] | 2 | No | No | No | No |
| | Shi et al. [37] | $N$ | No | No | Yes | No |
| | Kunzel et al. [35] | 2 | No | No | Yes | Yes |
| Learning across domains | Ganin et al. [38] | 2 | No | No | No | No |
| | Magliacane et al. [61] | N | No | No | No | No |
| | Rudner et al. [41] | N | No | No | Yes | Yes |
| | Yoon et al. [22] | N | Yes | No | Yes | Yes |
| | Moon et al. [60] | 2 | Yes | Yes | Yes | No |
| | (Ours) HTCE-learners | 2 | Yes | No | Yes | Yes |

Table 2: Comparison of our proposed method with related works.

# B   Heterogeneous transfer propensity estimator

For our *HTCE-propensity estimator* that is used as part of the two-step meta-learners (e.g. HTCE-DR-learner) we build a model that shares information between the treatment assignment mechanisms in the source and target datasets as illustrated in Figure 8. In particular, our HTCE-propensity estimator consists of encoders $\phi_\pi^{p_R}, \phi_\pi^{p_T}, \phi_\pi^s$ for the heterogeneous feature spaces, such that $\Phi_\pi^R(x^R) = [\phi_\pi^s(x^s)||\phi_\pi^{p_R}(x^R)]$ and $\Phi_\pi^T(x^T) = [\phi_\pi^s(x^s)||\phi_\pi^{p_T}(x^T)]$. To enable transfer, we use an approach similar to the one described in Section 4.2, but this time to share information between the hypothesis functions for the propensity estimation $\left(g_\pi^R(\Phi_\pi^R(x^R)), g_\pi^T(\Phi_\pi^T(x^T))\right)$. Moreover, we use the following outcome loss:

$$\mathcal{L}_\pi = \sum_{i=1}^{N_R} l(w_i, g_\pi^R(\Phi^R(x_i^R))) + \sum_{i=1}^{N_T} l(w_i, g_\pi^T(\Phi^T(x_i^T))) \tag{9}$$

where $l(\cdot, \cdot)$ represents the binary cross-entropy loss (for binary treatment).

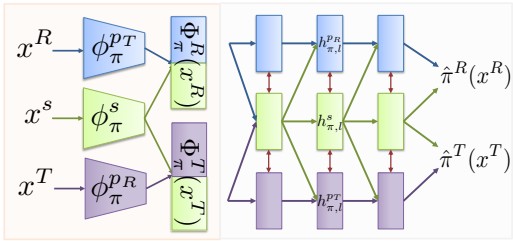

Figure 8: HTCE-propensity estimator.

# C   Pseudo-code for the HTCE-learners

Algorithm 1 provides the pseudo-code for training our proposed HTCE-T-learner and Algorithm 2 provides the pseudo-code for training our proposed HTCE-TARNet. Note that a similar training procedure can be used for the HTCE-S-Learner, HTCE-propensity estimator and in the second stage of the two-step learners (e.g. HTCE-DR-learner). The main differences are in the encoder used, in terms of sharing layers between PO functions within each domain and the outcome used for computing the corresponding loss function. Refer to Section 5.1 and Appendix B for architectural details. Similarly to [15], we do not use cross-fitting for the two-step learners (HTCE-DR-learner). Refer to [15] for more details. For simplicity, we set the mini-batch size to be the same in both the source and target datasets: $B_T = B_R = B$. Moreover, note that in practice, we multiply $\mathcal{L}_{\text{orth}_z}$ and $\mathcal{L}_{\text{orth}_{PO}}$ by $0.01$ to ensure that all losses have a similar scale and we use early stopping on the validation dataset from the target domain to check for convergence.

**Algorithm 1** Pseudo-code for training the HTCE-T-learner.

---

1: Input: Dataset with expert demonstrations from source $\mathcal{D}^R$ and target $\mathcal{D}^T$ domains, learning rate $\lambda$, mini-batch size for source dataset $B_R$, mini-batch size for target dataset $B_T$
2: Initialize: $\theta$ (all parameters of the HTCE-T-Learner)
3: **while** not converged **do**
4:     Sample mini-batch of $B_R$ demonstrations from the source dataset $(x_i^R, w_i, y_i) \sim \mathcal{D}^R$ and a minibatch of $B_T$ demonstrations from the target dataset $(x_i^T, w_i, y_i) \sim \mathcal{D}^T$.
5:     **Handle heterogeneous feature spaces**:
6:     **for** $i = 1 \ldots B_R$ **do**                             ▷ Process batch from source domain.
7:         $z_i^{s_R} = \phi_{w_i}^s(x_i^s)$, where $x_i^s$ is part of $x_i^R$; $z_i^{p_R} = \phi_{w_i}^R(x_i^R)$
8:     Set: $\zeta^{s_R} = [z_1^{s_R} \ldots z_{B_R}^{s_R}]^\top$, $\zeta^{p_R} = [z_1^{p_R} \ldots z_{B_R}^{p_R}]^\top$
9:     **for** $i = 1 \ldots B_T$ **do**                             ▷ Process batch from target domain.
10:         $z_i^{s_T} = \phi_{w_i}^s(x_i^s)$, where $x_i^s$ is part of $x_i^T$; $z_i^{p_T} = \phi_{w_i}^R(x_i^T)$
11:     Set: $\zeta^{s_T} = [z_1^{s_T} \ldots z_{B_T}^{s_T}]^\top$, $\zeta^{p_T} = [z_1^{p_T} \ldots z_{B_T}^{p_T}]^T$
12:     Compute orthogonal regularization loss for the feature representations:

$$\mathcal{L}_{\text{orth}_z} = \|\zeta^{s_R\,\top} \zeta^{p_R}\|_F^2 + \|\zeta^{s_T\,\top} \zeta^{p_T}\|_F^2$$

13:     **Share information between PO functions across domains**:
14:     **for** $i = 1 \ldots B_R$ **do**                             ▷ Process batch from source domain.
15:         **for** $l = 1 \ldots L$ **do**
16:             **if** $l == 1$ **then**
17:                 $\tilde{h}_{w_i,1}^{p_R} = [z_i^{s_R}||z_i^{p_R}]$; $\tilde{h}_{w_i,1}^s = \tilde{h}_{w_i,1}^{p_R}$
18:             **else**
19:                 $\tilde{h}_{w_i,l}^{p_R} = [h_{w_i,l-1}^s||h_{w_i,l-1}^{p_R}]$; $\tilde{h}_{w,l}^s = [h_{w,l-1}^s]$.
20:             $h_{w_i,l}^s = \text{Shared\_Layer}_{w_i,l}(\tilde{h}_{w_i,l}^s)$; $h_{w_i,l}^{p_R} = \text{Private\_Layer}_{w_i,l}^R(\tilde{h}_{w_i,l}^{p_R})$
21:         $\hat{y}_i^R = \psi(h_{w_i,L}^{p_R} + h_{w_i,L}^s)$
22:     **for** $i = 1 \ldots B_T$ **do**                             ▷ Process batch from target domain.
23:         **for** $l = 1 \ldots L$ **do**
24:             **if** $l == 1$ **then**
25:                 $\tilde{h}_{w_i,1}^{p_T} = [z_i^{s_T}||z_i^{p_T}]$; $\tilde{h}_{w_i,1}^s = \tilde{h}_{w_i,1}^{p_T}$
26:             **else**
27:                 $\tilde{h}_{w_i,l}^{p_T} = [h_{w_i,l-1}^s||h_{w_i,l-1}^{p_T}]$, $\tilde{h}_{w_i,l}^s = [h_{w_i,l-1}^s]$.
28:             $h_{w_i,l}^s = \text{Shared\_Layer}_{w_i,l}(\tilde{h}_{w_i,l}^s)$; $h_{w_i,l}^{p_T} = \text{Private\_Layer}_{w_i,l}^T(\tilde{h}_{w_i,l}^{p_T})$
29:         $\hat{y}_i^T = \psi(h_{w_i,L}^{p_T} + h_{w_i,L}^s)$
30:     Compute outcome loss:

$$\mathcal{L}_y = \sum_{i=1}^{B_R} l(y_i, \hat{y}_i^R) + \sum_{i=1}^{B_T} l(y_i, \hat{y}_i^T)$$

31:     Compute orthogonal regularization loss for the PO shared and private layers:

$$\mathcal{L}_{\text{orthPO}} = \sum_{w \in \{0,1\}} \sum_{l=1}^{L} \|\theta_{w,l}^{s\,\top} \theta_{w,l,1:m_{l-1}^s}^{p_R}\|_F^2 + \|\theta_{w,l}^{s\,\top} \theta_{w,l,1:m_{w,l-1}^s}^{p_T}\|_F^2$$

32:     Parameters update:
33:     $\theta \leftarrow \theta - \lambda \nabla_\theta (\mathcal{L}_y + \mathcal{L}_{\text{orth}_z} + \mathcal{L}_{\text{orthPO}})$
34: **Output:** Learnt parameters $\theta$

---

**Algorithm 2** Pseudo-code for training the HTCE-TARNet.

1: Input: Dataset with expert demonstrations from source $\mathcal{D}^R$ and target $\mathcal{D}^T$ domains, learning rate $\lambda$, mini-batch size for source dataset $B_R$, mini-batch size for target dataset $B_T$

2: Initialize: $\theta$ (all parameters of the HTCE-TARNet)

3: **while** not converged **do**

4:     Sample mini-batch of $B_R$ demonstrations from the source dataset $(x_i^R, w_i, y_i) \sim \mathcal{D}^R$ and a minibatch of $B_T$ demonstrations from the target dataset $(x_i^T, w_i, y_i) \sim \mathcal{D}^T$.

5:     **Handle heterogeneous feature spaces**:

6:     **for** $i = 1 \ldots B_R$ **do**                  ▷ Process batch from source domain.

7:         $z_i^{s_R} = \phi^s(x_i^s)$, where $x_i^s$ is part of $x_i^R$; $z_i^{p_R} = \phi^R(x_i^R)$

8:     Set: $\zeta^{s_R} = [z_1^{s_R} \ldots z_{B_R}^{s_R}]^\top$, $\zeta^{p_R} = [z_1^{p_R} \ldots z_{B_R}^{p_R}]^\top$

9:     **for** $i = 1 \ldots B_T$ **do**                  ▷ Process batch from target domain.

10:         $z_i^{s_T} = \phi^s(x_i^s)$, where $x_i^s$ is part of $x_i^T$; $z_i^{p_T} = \phi^R(x_i^T)$

11:     Set: $\zeta^{s_T} = [z_1^{s_T} \ldots z_{B_T}^{s_T}]^\top$, $\zeta^{p_T} = [z_1^{p_T} \ldots z_{B_T}^{p_T}]^T$

12:     Compute orthogonal regularization loss for the feature representation:

$$\mathcal{L}_{\text{orth}_z} = \|\zeta^{s_R \top} \zeta^{p_R}\|_F^2 + \|\zeta^{s_T \top} \zeta^{p_T}\|_F^2$$

13:     **Share information between PO functions within each domain**:

14:     **for** $i = 1 \ldots B_R$ **do**                  ▷ Process batch from source domain.

15:         $\tilde{z}_i^R = \Omega^{p_R}([z_i^{s_R} || z_i^{p_R}])$

16:     **for** $i = 1 \ldots B_T$ **do**                  ▷ Process batch from target domain.

17:         $\tilde{z}_i^T = \Omega^{p_T}([z_i^{s_T} || z_i^{p_T}])$

18:     **Share information between PO functions across domains**:

19:     **for** $i = 1 \ldots B_R$ **do**                  ▷ Process batch from source domain.

20:         **for** $l = 1 \ldots L$ **do**

21:             **if** $l == 1$ **then**

22:                 $\tilde{h}_{w_i,1}^{p_R} = \tilde{z}_i^R$; $\tilde{h}_{w_i,1}^s = \tilde{h}_{w_i,1}^{p_R}$

23:             **else**

24:                 $\tilde{h}_{w_i,l}^{p_R} = [h_{w_i,l-1}^s || h_{w_i,l-1}^{p_R}]$; $\tilde{h}_{w,l}^s = [h_{w,l-1}^s]$.

25:             $h_{w_i,l}^s = \text{Shared\_Layer}_{w_i,l}(\tilde{h}_{w_i,l}^s)$; $h_{w_i,l}^{p_R} = \text{Private\_Layer}_{w_i,l}^R(\tilde{h}_{w_i,l}^{p_R})$

26:         $\hat{y}_i^R = \psi(h_{w_i,L}^{p_R} + h_{w_i,L}^s)$

27:     **for** $i = 1 \ldots B_T$ **do**                  ▷ Process batch from target domain.

28:         **for** $l = 1 \ldots L$ **do**

29:             **if** $l == 1$ **then**

30:                 $\tilde{h}_{w_i,1}^{p_T} = \tilde{z}_i^T$; $\tilde{h}_{w_i,1}^s = \tilde{h}_{w_i,1}^{p_T}$

31:             **else**

32:                 $\tilde{h}_{w_i,l}^{p_T} = [h_{w_i,l-1}^s || h_{w_i,l-1}^{p_T}]$, $\tilde{h}_{w_i,l}^s = [h_{w_i,l-1}^s]$.

33:             $h_{w_i,l}^s = \text{Shared\_Layer}_{w_i,l}(\tilde{h}_{w_i,l}^s)$; $h_{w_i,l}^{p_T} = \text{Private\_Layer}_{w_i,l}^T(\tilde{h}_{w_i,l}^{p_T})$

34:         $\hat{y}_i^T = \psi(h_{w_i,L}^{p_T} + h_{w_i,L}^s)$

35:     Compute outcome loss:

$$\mathcal{L}_y = \sum_{i=1}^{B_R} l(y_i, \hat{y}_i^R) + \sum_{i=1}^{B_T} l(y_i, \hat{y}_i^T)$$

36:     Compute orthogonal regularization loss for the PO shared and private layers:

$$\mathcal{L}_{\text{orth}_{\text{PO}}} = \sum_{w \in \{0,1\}} \sum_{l=1}^{L} \|\theta_{w,l}^s{}^\top \theta_{w,l,1:m_{l-1}^s}^{p_R}\|_F^2 + \|\theta_{w,l}^s{}^\top \theta_{w,l,1:m_{w,l-1}^s}^{p_T}\|_F^2$$

37:     Parameters update:

38:     $\theta \leftarrow \theta - \lambda \nabla_\theta (\mathcal{L}_y + \mathcal{L}_{\text{orth}_z} + \mathcal{L}_{\text{orth}_{\text{PO}}})$

39: **Output:** Learnt parameters $\theta$

# D Experimental details

## D.1 Dataset description

In this section, we provide a description of the three datasets used for evaluation: MAGGIC [45], Twins [43] and TCGA [44]. Note that these datasets have diverse characteristics in terms of the number of features and the proportion of categorical/continuous features. To the best of our knowledge, none of the medical datasets used contain personally identifiable information nor offensive content. Given that Twins and TCGA are datasets from a single domain, we then explain how we used them to obtain source and target datasets with heterogeneous feature spaces.

**MAGGIC.** MAGGIC [45] consists of datasets with heterogeneous features from 30 domains representing medical studies of patients who have experienced heart failure. Note that this medical dataset is private. Refer to [45] for details on how the dataset was collected and curated. Each medical study consists of a number of patients ranging from 190 to 13279. From these, we sample a dataset with $< 500$ patients to represent our target domain and a dataset with $> 500$ patients to represent our source domain. The different datasets in MAGGIC consist of a mixture of categorical and continuous features representing clinical covariates for patients who have experienced heart failure such as age, gender, serum creatinine, diabetes, beta-blocker prescription, lower systolic blood pressure, lower body mass, time since diagnosis, current smoker, chronic obstructive pulmonary disease, etc. The total number of features across the different medical studies in MAGGIC is 216 and the average number of features in a single study is 73. After sampling the source and target dataset from the different medical studies in MAGGIC, we simulate treatment assignments and outcomes as described in Section 6. Note that different source and target datasets are sampled for each of the different 10 random seeds used for all experimental results.

**Twins.** Twins [43] represents one of the standard benchmark datasets in the causal inference literature for estimating the effects of binary treatments. The dataset consists of data from 11400 pairs of twin births in the USA recorded between 1989-1991 [43]. Refer to [43] for details of how the dataset was collected and curated. We use the publicly available version of the dataset from: `https://github.com/AliciaCurth/CATENets`. For each twin pair, 39 relevant covariates were recorded related to the parents, pregnancy and birth such as marital status, mother's age, number of previous births, pregnancy risk factors, quality of care during pregnancy, number of gestation weeks, etc. These represent a mixture of continuous and categorical features. We obtain our full dataset with $N_F = 114000$ examples and $D_F = 39$ features by randomly sampling one of the twins to observe. We then sub-sample our source and target datasets as described below and simulate treatments and outcomes as described in Section 6.

**TCGA.** TCGA is a dataset consisting of cancer patients for which gene expression measurements were recorded [44]. The dataset is publicly available and has been used by previous methods in causal inference [46, 47, 62]. In particular, we use the publicly available version of the TCGA dataset as used by DRNet [47] `https://github.com/d909b/drnet`. Refer to [44] for details of how the dataset has been collected and curated. The dataset consists of $N_F = 9659$ patients (samples), for which we use the measurements from the $D_F = 100$ most variable genes. Note that these are all continuous features. Moreover, we log-normalized the gene expression data and scale each feature in the $[0, 1]$ interval. We then sub-sample our source and target datasets as described below. The treatment assignment and outcomes are simulated as described in Section 6.

**Obtaining source and target datasets for Twins and TCGA.** Unless the size of the target dataset is fixed, from the full datasets, we sample a target dataset of size $N_T \sim \mathcal{U}(100, 500)$ and a source dataset of size $N_R \sim \mathcal{U}(1000, N_F - N_T)$, where $N_T$ is the size of the target dataset, $N_R$ is the size of the source dataset and $N_F$ is the size of the full dataset. For the experiments where we vary the size of the target dataset, we set $N_T \in \{100, 200, 300, 500, 1000, 2000, 4000\}$ and we again sample the size of the source dataset from $N_R \sim \mathcal{U}(1000, N_F - N_T)$. Moreover, to create heterogeneous feature spaces for the source and target domains, let $D_F$ be the number of features in the full dataset. From these, we randomly sample $D_{p_R} \in \mathcal{U}(5, D_F/3)$ features that are private for the source dataset, $D_{p_T} \in \mathcal{U}(5, D_F/3)$ features that are private for the target dataset and $D_s \in \mathcal{U}(5, D_F/3)$ features that are shared between the two. Note that different sizes for the source and target datasets and different sets of shared and private features are sampled for each of the different 10 random seeds used for all experimental results.

We use the full source datasets for training the benchmarks making use of the source dataset. Each target dataset undergoes a split into 56/24/20% for training, validation and testing respectively. The validation dataset is used for early stopping.

**Summary statistics for the datasets** We provide in Table 3 summary statistics of the different characteristics of the full source and target datasets used across our 10 experimental runs. Note that for the experiments where we vary the size of the target dataset, $N_T$ is fixed as described above. To highlight the diversity of the datasets used for evaluating the benchmarks, we highlight in Table 3 the mean and standard deviations for the size of the source dataset $N_R$, size of the target dataset $N_T$, number of features shared between the source and target datasets $D_s$, number of features private to the source dataset $D_{p_R}$, number of features private to the target dataset $D_{p_T}$, proportion of shared features in the source dataset $D_s/(D_s + D_{p_R})$, proportion of shared features in the target dataset $D_s/(D_s + D_{p_T})$ and the Maximum Mean Discrepancy (MMD) (computed using RBF kernel) between the shared features of the source and target datasets $\mathrm{MMD}(X^{s_R}, X^{s_T})$. Note that the MMD between the shared features of the source and target datasets for MAGGIC is much higher than for TCGA and Twins, as the different datasets in MAGGIC represent medical studies with different patient populations.

Table 3: Summary statistics of the datasets sampled for training the benchmarks.

|  | TCGA | Twins | MAGGIC |
|---|---|---|---|
| $N_R$ | $4219 \pm 2569$ | $5106 \pm 2988$ | $2533 \pm 3711$ |
| $N_T$ | $360 \pm 120$ | $360 \pm 120$ | $267 \pm 75$ |
| $D_s$ | $20 \pm 8$ | $9 \pm 2$ | $27 \pm 7$ |
| $D_{p_R}$ | $15 \pm 7$ | $9 \pm 2$ | $25 \pm 9$ |
| $D_{p_T}$ | $21 \pm 9$ | $8 \pm 3$ | $21 \pm 15$ |
| $D_s/(D_s + D_{p_R})$ | $0.57 \pm 0.17$ | $0.49 \pm 0.10$ | $0.54 \pm 0.13$ |
| $D_s/(D_s + D_{p_T})$ | $0.50 \pm 0.14$ | $0.52 \pm 0.08$ | $0.60 \pm 0.19$ |
| $\mathrm{MMD}(X^{s_R}, X^{s_T})$ | $0.0009 \pm 0.0005$ | $0.0011 \pm 0.0008$ | $0.056 \pm 0.057$ |

### D.2 Implementation details and hyperparameter setting

As mentioned in Section 6, we evaluate the following benchmarks (1) training the CATE-learners only on the target dataset, (2) using only the shared features between the source and target datasets and (3) using RadialGAN [22] to 'translate' the source dataset into the target domain (4) training the HTCE-learners on the full source and target datasets. We describe here the implementation details for RadialGAN [22], for the different CATE-learners used (S-Learner, T-Learner, DR-Learner and TARNet) and the corresponding HTCE-learners (HTCE-S-Learner, HTCE-T-Learner, HTCE-DR-Learner and HTCE-TARNet).

**RadialGAN.** We use the publicly available implementation of RadialGAN [22][1] which we augment to the treatment effects setting by considering the treatment separate from the rest of the features. While RadialGAN, was developed to work with any number of source datasets and uses multiple GAN architectures to learn to translate from any one of the source datasets to the target domain, we consider here a simplified case with one source dataset. We use the same hyperparameter setting described in [22] and found in the publicly available implementation for the generators, discriminators, encoders and decoders architectures and optimization (optimizer, batch size, learning rate).

**CATE-learners.** We use the publicly available implementation of the different CATE learners from [15][2]. This allows us to have a consistent implementation for the S-Learner, T-Learner, DR-Learner and TARNet [10]. Moreover, we use components similar to those in [10, 15] for all networks which ensure that $\hat{\mu}_w(x)$, $\hat{\pi}(x)$ and $\hat{\tau}(x)$ have access to similar number of layers and units in total, thus resulting in equally complex nuisance functions. In particular, for the S-learner, $\hat{\mu}(x, w)$ consists of 5 layers of 200 units each, for the T-learner, each of $\hat{\mu}_0(x), \hat{\mu}_1(x)$ consists of 5 layers with 200 units each. For the DR-Learner, we set each of $\hat{\mu}_0(x), \hat{\mu}_1(x), \hat{\pi}(x), \hat{\tau}(x)$ to have 5 layers with 200 units each. Finally, TARNet consists of 3 representation layers of 200 units each, followed by two outcome heads, each with 2 layers of 100 units each. We use ReLU activation function for the hidden layers

---

[1] https://github.com/vanderschaarlab/mlforhealthlabpub/tree/main/alg/RadialGAN
[2] https://github.com/AliciaCurth/CATENets

and linear/sigmoid activation for the last layer (for continuous/binary outcomes respectively). The different CATE learners are trained using the Adam optimizer [63], with learning rate set to $0.0001$ and batch size $128$. We perform early stopping on the validation dataset from the target domain.

**HTCE-learners.** Each $\phi^{p_R}, \phi^{p_T}, \phi^s$ for handling the heterogeneous feature spaces consists of one hidden layer with 100 units each. For the HTCE-S-Learner, we consider a building block of 5 layers with 100 hidden units for each shared and private subspace to share information between $\hat{\mu}^R(x^R, w), \hat{\mu}^T(x^T, w)$. Note that this essentially gives us 200 hidden units for the hypothesis function in each domain, thus ensuring that the nuisance functions for the CATE vs. HTCE-learners in each domain have similar complexity. For the HTCE-T-learner, we use separate feature encoders for each treatment $w \in \{0, 1\}$ ($\phi_w^{p_R}, \phi_w^{p_T}, \phi_w^s$ each consisting of one hidden layer with 100 units), and we use two building blocks of 5 layers with 100 units for each shared and private subspace to share information between $(\hat{\mu}_0^R(x^R), \hat{\mu}_0^T(x^T))$ and $(\hat{\mu}_1^R(x^R), \hat{\mu}_1^T(x^T))$. Similarly, for the HTCE-DR-learner, we use the same architecture as for the HTCE-T-learner to obtain the plug-in estimates of the potential outcomes, which consists of separate feature encoders for each treatment $w$, and two building blocks of 5 layers with 100 units for each shared and private subspace to share information between $(\hat{\mu}_0^R(x^R), \hat{\mu}_0^T(x^T))$ and $(\hat{\mu}_1^R(x^R), \hat{\mu}_1^T(x^T))$. Moreover, for the propensity estimation, we use separate feature encoders ($\phi_\pi^{p_R}, \phi_\pi^{p_T}, \phi_\pi^s$ each consisting of one hidden layer with 100 units) and a building blocks of 5 layers with 100 units for each shared and private subspace to share information between $(\hat{\pi}^R(x^R), \hat{\pi}^T(x^T))$. Finally, for the pseudo-outcome regressor we also use separate feature encoders ($\phi_y^{p_R}, \phi_y^{p_T}, \phi_y^s$ each consisting of one hidden layer with 100 units) and a building blocks of 5 layers with 100 units for each shared and private subspace to share information between $(\hat{\tau}^R(x^R), \hat{\tau}^T(x^T))$. For HTCE-TARNet, we use feature encoders shared between the two treatments ($\phi^{p_R}, \phi^{p_T}, \phi^s$ each consisting of one hidden layer with 100 units), followed by 3 representation layers (for each of $\Omega^R$, $\Omega^T$) and two building blocks of 2 layers with 100 units for each shared and private subspace to share information between $(\hat{\mu}_0^R(x^R), \hat{\mu}_0^T(x^T))$ and $(\hat{\mu}_1^R(x^R), \hat{\mu}_1^T(x^T))$. We use ReLU activation function for the hidden representation layers, SeLU activation for the shared and private subspace layers and linear/sigmoid activation for the last layer (for continuous/binary outcomes respectively). Note that for the building block of sharing information between outcome functions, we build upon the FlexTENet implementation [14]. The different HTCE-learners are trained using the Adam optimizer [63], with learning rate set to $0.0001$ and batch size $128$. Moreover, we also use the validation dataset from the target domain for early stopping.

All of the experiments were run on a virtual machine with 6CPUs, an Nvidia K80 Tesla GPU and 56GB of RAM.

# E   Additional experimental results

This section provides results for the different experimental settings considered in Section 6 for the remaining datasets and for all learners. Note that we observe similar trends for the different experimental settings across all datasets.

## E.1   Benchmarks comparison and source of gain

Table 4 reports results on the TCGA and MAGGIC datasets for the comparison of the HTCE-learners against the benchmarks and the source of gain analysis. To reiterate, we compare the following benchmarks (1) training the CATE learners only on the target dataset, (2) using only the shared features between the source and target datasets to train the CATE learners (3) using RadialGAN [22] to 'translate' the source dataset into the target domain and training the CATE learners on the target dataset augmented in this way and (4) training our HTCE-learners on the full source and target datasets. Moreover, we evaluate the impact of removing the shared and private layers that enable sharing information between the PO functions across domains (HTCE - PO sharing), removing the orthogonal regularization loss for the learnt shared and private feature representations in Equation 3 (HTCE - $\mathcal{L}_{\mathrm{orth}_z}$) and removing orthogonal regularization loss from the PO layers in Equation 5 (HTCE - $\mathcal{L}_{\mathrm{ortho}_{\mathrm{PO}}}$). We note that on the TCGA and MAGGIC datasets as well, the HTCE-learners achieve better performance compared to the baselines and that each component we propose brings performance improvements.

Table 4: Benchmarks comparison and source of gain analysis in terms of PEHE on TCGA and MAGGIC datasets.

| Learner | TCGA | | | | MAGGIC | | | |
|---|---|---|---|---|---|---|---|---|
| | S-Learner | T-Learner | DR-Learner | TARNet | S-Learner | T-Learner | DR-Learner | TARNet |
| Target | $0.15 \pm 0.02$ | $0.12 \pm 0.01$ | $0.10 \pm 0.01$ | $0.11 \pm 0.01$ | $0.39 \pm 0.04$ | $0.21 \pm 0.01$ | $0.19 \pm 0.01$ | $0.19 \pm 0.01$ |
| Shared features | $0.30 \pm 0.07$ | $0.30 \pm 0.07$ | $0.29 \pm 0.07$ | $0.30 \pm 0.07$ | $0.32 \pm 0.03$ | $0.28 \pm 0.03$ | $0.27 \pm 0.03$ | $0.27 \pm 0.03$ |
| RadialGAN | $0.12 \pm 0.01$ | $0.11 \pm 0.01$ | $0.09 \pm 0.01$ | $0.13 \pm 0.03$ | $0.31 \pm 0.01$ | $0.19 \pm 0.01$ | $0.19 \pm 0.02$ | $0.19 \pm 0.01$ |
| HTCE - PO sharing | $0.15 \pm 0.03$ | $0.10 \pm 0.01$ | $0.09 \pm 0.01$ | $0.10 \pm 0.01$ | $0.29 \pm 0.02$ | $0.15 \pm 0.01$ | $0.14 \pm 0.01$ | $0.17 \pm 0.01$ |
| HTCE - $\mathcal{L}_{\mathrm{orth}_z}$ | $0.10 \pm 0.01$ | $0.07 \pm 0.01$ | $0.06 \pm 0.01$ | $0.07 \pm 0.01$ | $0.25 \pm 0.02$ | $0.10 \pm 0.01$ | $0.09 \pm 0.01$ | $0.12 \pm 0.01$ |
| HTCE - $\mathcal{L}_{\mathrm{ortho}_{\mathrm{PO}}}$ | $0.11 \pm 0.01$ | $0.08 \pm 0.01$ | $0.07 \pm 0.01$ | $0.05 \pm 0.01$ | $0.26 \pm 0.02$ | $0.10 \pm 0.01$ | $0.10 \pm 0.01$ | $0.12 \pm 0.01$ |
| HTCE (ours) | $\mathbf{0.07} \pm 0.01$ | $\mathbf{0.06} \pm 0.01$ | $\mathbf{0.04} \pm 0.01$ | $\mathbf{0.06} \pm 0.01$ | $\mathbf{0.24} \pm 0.02$ | $\mathbf{0.08} \pm 0.01$ | $\mathbf{0.08} \pm 0.01$ | $\mathbf{0.10} \pm 0.01$ |

When looking at the performance of the benchmarks, using only the shared features between the two datasets generally has poor performance. This is due to the fact that this introduces hidden confounders in the CATE estimation since all of the patient features in both domains are used to obtain the potential outcomes and the treatment assignment in our data simulation (see Section 6). Moreover, note that RadialGAN only slightly improves performance over only training the CATE learners on the target dataset. We hypothesise that this may also be due to the small size of the target dataset which hinders the generator networks in RadialGAN to learn how to map examples from the source domain to the target one.

## E.2   Varying the information sharing between domains

Figure 9 reports the results for varying the parameter $\alpha$, which controls the amount of information shared between the PO in the source and target domains, on the TCGA (top) and MAGGIC (bottom) datasets. We again notice that our HTCE-learners, which use a flexible approach for information sharing between PO functions across domains achieve good performance both when the PO functions are significantly different between the source and target dataset ($\alpha = 0.1$) and also when they have the same functional form and only depend on the shared features ($\alpha = 1.0$). In addition, for high values of $\alpha$ and especially for $\alpha = 1.0$ we again notice the degradation in performance due to the fact that the learners need to perform the implicit task of feature selection.

## E.3   Varying the target dataset size.

Figure 10 reports the results on when varying the size of the target dataset $N_T$ on the TCGA dataset. Note that for the MAGGIC dataset, the size of the target domain is not simulated and it represents the size of the various studies selected as target domains. Thus, we only report here results on TCGA where we directly simulate the size of the target dataset. We notice that the benefits of doing transfer learning degrade as we increase the size of the target dataset.

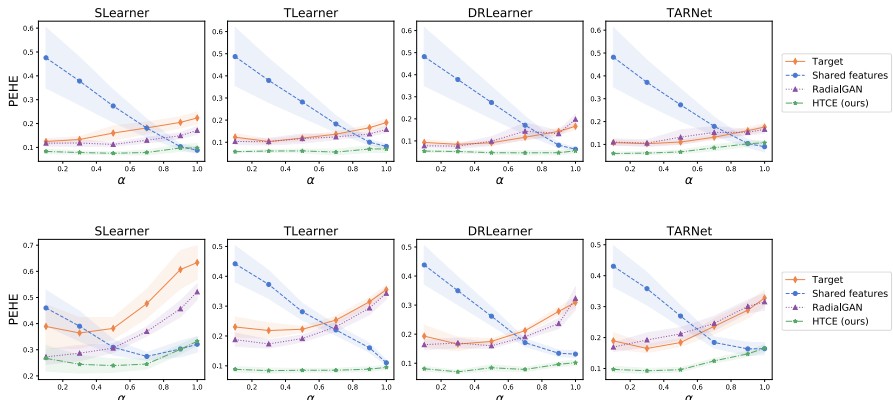

Figure 9: Performance comparison on TCGA (top) and MAGGIC (bottom) when varying the information sharing between PO functions across domains through $\alpha$.

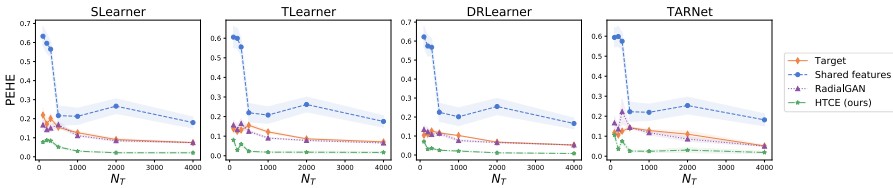

Figure 10: Performance comparison on TCGA when varying the the size of the target dataset ($N_T$).

### E.4 Effect of selection bias

Finally, we report results when varying the selection bias in the source and target domains for all learners on all three datasets: Twins (Figure 11), TCGA (Figure 12) and MAGGIC (Figure 13). We consider three setting for the selection bias in the source dataset $\kappa_R = 0.0$ for random treatment assignment, $\kappa_R = 2.0$ for moderate and $\kappa_R = 10.0$ for strong selection bias. For each setting of $\kappa_R$, we vary the selection bias in the target dataset $\kappa_T$ from $\kappa_T = 0.0$ to $\kappa_T = 10.0$. Our HTCE-learners have consistent performance when increasing the selection bias in both datasets and when there are significant discrepancies in the treatment assignment mechanism between the source and target datasets. While the impact of increasing the selection bias differs among the learners (e.g. TARNet vs. DRLearner), note that this is due to the intrinsic characteristics of each learner on a single domain, as the DRLearner uses propensity weighting to adjust for the selection bias, while TARNet does not.

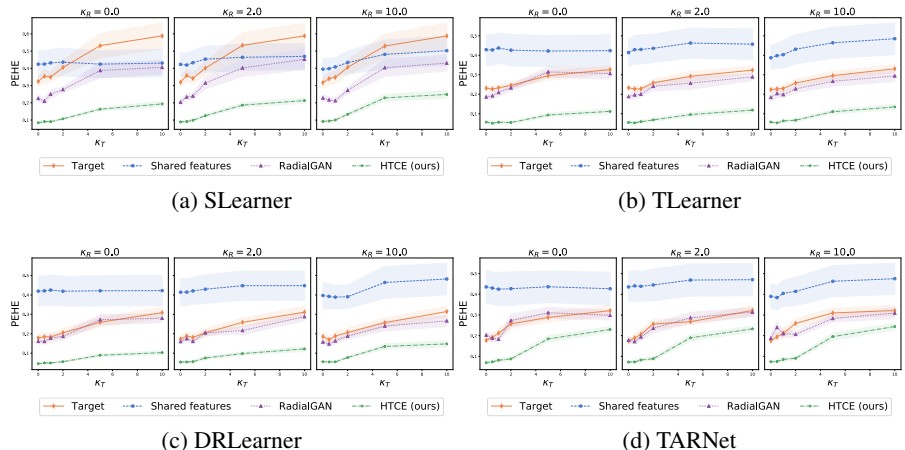

Figure 11: Performance comparison on Twins for all learners when varying the selection bias $\kappa_R$ and $\kappa_T$ in the source and target datasets.

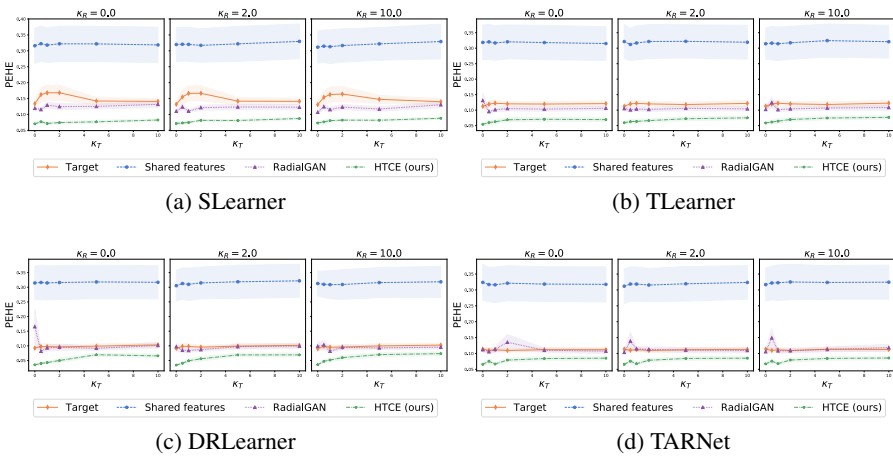

Figure 12: Performance comparison on TCGA for all learners when varying the selection bias $\kappa_R$ and $\kappa_T$ in the source and target datasets.

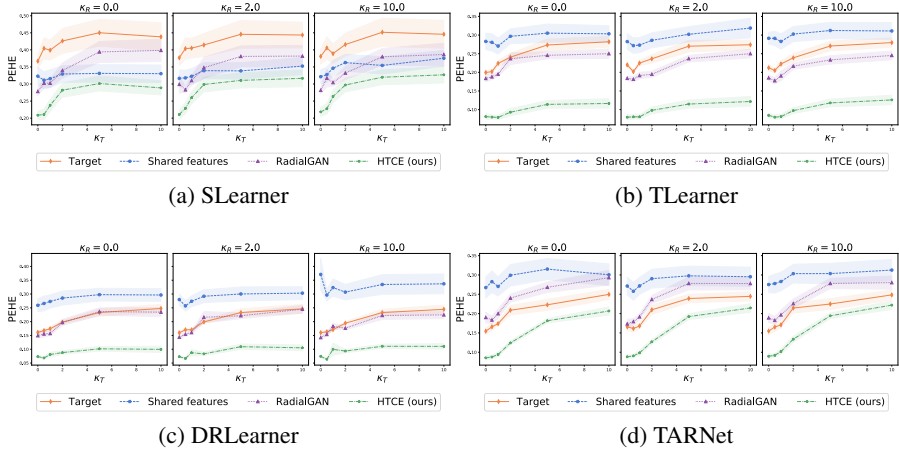

Figure 13: Performance comparison on MAGGIC for all learners when varying the selection bias $\kappa_R$ and $\kappa_T$ in the source and target datasets.

# F Additional discussion about extensions and limitations

In this section, we describe how our method could be extended to additional scenarios than the ones considered in the paper and further discuss limitations.

**Multiple source datasets.** While in this paper we mainly consider the standard and most common transfer learning setting [64] of leveraging a source dataset to improve the estimation of outcomes on a target dataset of interest, our proposed approach could be easily extended to having multiple source domains, and in fact, would scale linearly to incorporating additional domains.

More precisely, consider access to $M$ source domains, $\{\mathcal{D}^{R_m}\}_{m=1}^{M}$, where each source domain consists of $\mathcal{D}^{R_m} = \{(X_i^{R_m}, W_i, Y_i)\}_{i=1}^{N_{R_m}}$ and a target domain as described in Section 3. The objective would still be the one from Section 3 of improving the estimation of CATE for the target domain, but in this case, we need to leverage data from all $M$ source domains. Our proposed building blocks could be extended as follows to handle the additional source domains. The building block for handling heterogeneous feature spaces between source and target domains could be extended by having encoders $\phi^{R_m}$ for the private features $X^{p_{R_m}}$ of each source domain (part of $X^{R_m}$). Moreover, we can build an approach that shares information between the potential outcomes of each source domain with the target domain by having $L$ layers shared between all source domains and the target domains and $L$ private layers for each source domain. For source domain $m$, the input to the layer $(l + 1)$-th can be obtained using $\tilde{h}_{w,l+1}^{p_{R_m}} = [h_{w,l}^s || h_{w,l}^{p_{R_m}}]$ where $h_{w,l}^{p_{R_m}}$ is the output of the $l$-th private layer for source domain $m$ and $h_{w,l}^s$ is the output of the $l$-th shared layer across all domains.

While from a model development perspective this extension can be easily done, one also needs to consider whether the multiple source domains satisfy the underlying implicit assumptions for which such an architecture would be appropriate. In particular, it would be important to consider whether the PO functions across all source domains share information with the PO functions in the target domain, as using source domains that are significantly different from the target domain could harm performance.

**Streaming datasets.** Another important setting to consider is the one of having streaming datasets. In this scenario, one option could be the case where we already have a source domain and a target domain and we have incoming data streaming from the target domain. One such example in healthcare would be the case where in a single hospital we start collecting more or different clinical covariates for the patients which we now want to use for CATE estimation. The source domain would be the patients with only the original set of features, while the target domain would be the patients with the new set of features. However, as we start collecting these additional features, the initial target dataset will be small but it can start increasing with time as we observe more patients. In this setting, we can train an HTCE-learner with the initial data available from the source and target domains. However, instead of retraining the full HTCE-learner as we obtain more patients from the target domain, one option would be to fine-tune the weights using the incoming examples. While this is outside the scope of our paper, we believe that it would be important to investigate appropriate ways for performing such fine-tuning.

Another option could be the case of having full source datasets streaming, while the target dataset remains fixed. This would happen in the setting where for instance we gradually get access to data from multiple locations and we want to use these datasets as source datasets. In this scenario, one possibility would be to use the approach described above for having multiple source datasets and retraining a new model that incorporates all of the available source datasets as we get access to them. Another possibility would be to, instead of retraining a full model as we go from $M$ to $M + 1$ source domains, we can add the needed private encoder and layers for the $(M + 1)$-th domain and train only the new parameters and fine-tune the shared ones with the data from the new domain. While this is again outside our scope, it can provide interesting avenues for future work.

**Unknown domains.** In this paper, we assume that the source and target domains are known. However, given that we handle the setting of transfer learning for heterogeneous feature spaces, if the domains are unknown, one way to split available data into different domains, in this case, would be by using the same features to denote a single domain. For instance, if different patients in the dataset have recorded different sets of features, then the patients can be grouped according to having the same information collected for them and these can denote the different domains. Then, one needs to decide which represents the target domain, while keeping in mind that doing transfer learning is most

beneficial when the target dataset is small (as highlighted by our experimental results in Section 6.2 and Figure 6 (bottom)).

**Personalized feature spaces.** In the case where different features are available for different patients and the source and target domains are unknown, then the patients can be grouped according to their feature spaces as described in the paragraph above. On the other hand, if the source and target datasets are pre-defined and the patients within each dataset have different features available for them, one possible option would be to consider the super-set of their features as the different feature spaces and consider the features that are not available for each patient as missing. However, as our HTCE-learners has not been designed for this particular setting of having missing data, one would also need to investigate if additional assumptions (in addition to ensuring that the no hidden confounders and overlap assumptions are still satisfied) are needed to be able to obtain valid estimates of causal effects.