# OpenReview forum: "Transfer Learning on Heterogeneous Feature Spaces for Treatment Effects Estimation"
_NeurIPS.cc/2022/Conference — NeurIPS 2022 Accept_

### Official Review · Reviewer_6zHZ · 2022-07-08

**Rating:** 6
**Confidence:** 5
**Soundness:** 2 fair
**Presentation:** 4 excellent
**Contribution:** 3 good

**Summary:**

This paper proposes a Heterogeneous Transfer Causal Effect (HTCE) framework to improve treatment effect estimations on the target dataset under the heterogeneous transfer learning problem.

**Questions:**

1. In lines 110-111, you mentioned that [32] do not assume access to label information. But in my view, I think that should be the true definition of "transfer learning". In general, your problem is similar to shared machine learning.

2. In lines 230-231, do you mean $||$ instead of $+$?

3. Your figures 3.a and 3.b are inconsistent with the text. It is just a small mistake.

4. I think the most interesting parts are i) solving the problem when the target dataset is substantially different from the source dataset, especially when the target dataset is in a small sample size; ii) combining HTCE with different learners. But your experiments do not show the importance and significance. For both your simulated and benchmark datasets, the distributions of source and target datasets are very similar. Specifically, the DGP or the empirical distribution is fixed, and the only difference is that you split them into source and target. It is undoubted that involving shared information will give better results than losing shared information. That is inconsistent with your claim in lines 149-152. I suggest you show how small the sample size of the target dataset is, and how big the differences between the distributions of source and target datasets are before reporting your experimental results.

5. In your Appendix, there are some typos, e.g, there exists "size size". Also, is twins 11400 or 114000?

**Limitations:**

The limitations are well dicusessed.

**Strengths And Weaknesses:**

Originality: 3. This problem has already been noticed by some works. The basic structure of HTCE is actually the combination of layer sharing + FlexTENet.

Quality: 4. I think it is good when authors combine HTCE structure with meta-learners and TARNet. They are explicit.

Clarity: 5. This paper is well written. Section 4 is especially easy to follow. Good!

Significance: 3. I think the argument and setting of estimating TEs for heterogeneous transfer learning is meaningful, but I don't think the experiment parts show the significance of the proposed method aiming to this problem.

---

> ### Author Response · Authors · 2022-08-02
> **Response to Reviewer 6zHZ [Part 3/3]**
>
> (cont'd)
>
> |  Statistic           | TCGA | Twins     | MAGGIC |
> | :---       |    :----: |     :----: |         :----: |
> | $N_R$    | $4219 \pm 2569$ |  $5106 \pm 2988$  | $2533 \pm 3711$  |
> | $N_T$   | $360 \pm 120$ |   $360 \pm 120$ | $267 \pm 75$  |
> | $D_s$    | $20 \pm 8$ | $9 \pm 2$ |  $27 \pm 7$ |
> | $D_{p_R}$   | $15 \pm 7$ |  $9 \pm 2$ |   $25 \pm 9$ |
> | $D_{p_T}$    | $21 \pm 9$ |   $8 \pm 3$  | $21 \pm 15$ |
> | $D_s / (D_s + D_{p_R})$   | $0.57 \pm 0.17$ |  $0.49 \pm 0.10$  |   $0.54 \pm 0.13$  |
> |$D_s / (D_s + D_{p_T})$| $0.50 \pm 0.14$  | $0.52 \pm 0.08$ | $0.60 \pm 0.19$|
> |$\text{MMD}(X^{s_R}, X^{s_T})$ | $0.0009 \pm 0.0005$ |  $0.0011 \pm 0.0008$ |  $0.056 \pm 0.057$ |
>
> Moreover, we would like to re-emphasize that these real datasets are only used to obtain the patient characteristics and, as descibed in in Section 6.1 we propose a new semi-synthetic data simulation for the outcomes and treatment assignments to evaluate the benchmarks in our heterogenous transfer learning set-up. Thus, please note that the data generating process is **not** the same for both the source and target datasets. In particular, once we obtain the heterogeneous features from the real-dataset, we use **different** functions to generate the potential outcomes in the source and target domains. As described in equations (7) and (8), the function for generating the potential outcomes (Y(w)^{R}) in the source domain uses the parameters $v\_{w, j}^{s}, v\_{w, j}^{p\_{R}}, v\_{j}^{{R}}$, while the function for generating the potential outcomes in the target domain (Y(w)^{R}) uses the parameters $v\_{w, j}^{s}, v\_{w, j}^{p\_{T}}, v\_{j}^{{T}}$. Out of these, $v\_{w, j}^{s}$ are the same in both domains, while the others are sampled differently, thus resulting in different outcome distributions $p(Y(w) \mid X^{R}) \neq p(Y(w) \mid X^{T})$. Note also that the potential outcomes in each domain depend on the private features characteristics for each domain. In addition given that the treatments are assigned based on the difference in PO in each domain $W\mid X^R \sim \text{Bernoulli}(\text{Sigmoid}(\kappa (Y^R(1) - Y^R(0))))$ and  $W\mid X^T \sim \text{Bernoulli}(\text{Sigmoid}(\kappa (Y^T(1) - Y^T(0))))$, we also have that $p(W=1 \mid X^{R}) \neq p(W=1 \mid X^{T})$. The full observational datasets for the source and target domains used for training are obtained by using the patient features sampled from the real-datasets, the simulated treatments and the corresponding simulated potential outcome for the simulated treatment. For testing, we evaluate the ability of the benchmarks to predict the difference in potential outcomes for the test split of the target dataset.
>
>
>
> Please also note that in our experiments, we also investigate
> * (1) performance when varying the size of the target dataset (see Figure 6 (bottom) and Section 6.2) and we indeed note that our method brings most benefits when the target dataset is small
> * (2) performance when varying the amount of information shared between the PO in the source and target datase through the parameters $\alpha$ (this denotes how much the potential outcomes depend on the shared features as opposed to the private features). Note that when $\alpha$ is low, the PO depend more on the private features, thus making their conditional distributions even more different across the domains. See Figure 6 (bottom) and Section 6.2 for the discussion of results.
>
> Due to space limitations for the revised person in the rebuttal period, it is not possible to include many experimental details in the main paper which is why we have updated Appendix D with the additional dataset details. However, we will move the details about how the shared and private features are samples and about the differences in datasets in the camera-ready version of the paper (which allows for an extra page).
>
> ### 5. Typos
>
> We have fixed the typo of ‘size size’. Moreover, the Twins dataset does indeed only have 11400 examples and not 114000. We have also fixed this typo in the revised manuscript.
>
>
> —------------------------
>
>
> We thank the reviewer again for the insightful comments! Please let us know if you have any other concerns as we are eager to address them.
>
>
> [R1] Pan, Sinno Jialin, and Qiang Yang. "A survey on transfer learning." IEEE Transactions on knowledge and data engineering 22, no. 10 (2010): 1345-1359.

---

> > ### Comment · Reviewer_6zHZ · 2022-08-03
> > **Thank you for the detailed responses**
> >
> > My concerns are all addressed. I would remain the score.

---

> ### Author Response · Authors · 2022-08-02
> **Response to Reviewer 6zHZ [Part 2/3]**
>
>
>
> ### 4. Experiments and clarifications of differences between source and target domains
>
> Please allows us to first clarify what we consider as differences between source and target domains. We consider that the features spaces of the source and target domains are heterogeneous (which is why we write $p(X^{R}) \neq p(X^{T}))$, but not completely disjoint (note that we have now slightly revised lines L151-155 to incorporate this clarification). As mentioned on lines L181-188, we consider that the source and target covariates can be split into $X^R = (X^s, X^{p_R} ) $ and $X^T=(X^s, X^{p\_T}) $ such that we have a set of features private (specific) to the source dataset $X^{p_R} \in \mathbb{R}^{D_{p_R}}$, a set of features private to the target dataset $X^{p_T} \in \mathbb{R}^{D_{p_T}}$ and a set of shared features between the two datasets $X^{s} \in \mathbb{R}^{D_S}$.
>
> In addition, in accordance with the definition of transfer learning [R1], we consider that the conditional outcome distribution is different $p(Y(w) \mid X^{R}) \neq p(Y(w) \mid X^{T})$ between the source and target domains. Moreover, to allow for different degrees of selection bias to be present in the source and target datasets we also consider that $p(W=1 \mid X^{R}) \neq p(W=1 \mid X^{T})$ (thus improving the generality of the method).
>
> Please allows us to now explain how for the experiments, we have used datasets that satisfy these requirements and assumptions. To begin with, we only obtain $X^R$ and $X^T$ from the following real datasets: TCGA, Twins and MAGGIC as described in Appendix D. The treatment assignments and outcomes are simulated as described in Section 6.1.
>
> MAGGIC consists of datasets with heterogeneous features from 30 domains representing medical studies of patients who have experienced heart failure. During training, we obtain the features for the target dataset $X^T$ by taking the patient characteristics from a randomly sampled dataset with $<500$ patients, while the features for the source dataset represent the patient characteristics from a randomly sampled dataset with  $>500$ patients (from the 30 medical studies in MAGGIC). Alternatively, given that Twins and TCGA originally represent a single dataset (for which we only consider the patient features) we randomly subsample parts of them to obtain the features for the source and target domains. In particular, for both Twins and TCGA  we sample features for the target dataset to be of size $N_T \sim \mathcal{U}(100, 500)$ and a source dataset of size $N_R \sim \mathcal{U}(1000, N_F-N_T)$, where $N_T$ is the size of the target dataset, $N_R$ is the size of the source dataset and $N_F$ is the size of the full dataset. Moreover, to create heterogeneous feature spaces for the source and target domains, let $D_F$ be the number of features in each full dataset (for Twins and TCGA). From these, we randomly sample $D_{p_R} \in \mathcal{U}(5, D_{F}/3)$ features that are private for the source dataset, $D_{p_T} \in \mathcal{U}(5, D_{F}/3)$ features that are private for the target dataset and $D_{s} \in \mathcal{U}(5, D_{F}/3)$ features that are shared between the two.
>
> For all of TCGA, Twins and MAGGIC, the different features for the source and target datasets are re-sampled for each of the different 10 random seeds used for all experimental results. In the following table, we provide summary statistics of the different characteristics of the patient features in the full source and target datasets used across our 10 experimental runs. Note that for the experiments where we vary the size of the target dataset, $N_T$ is fixed. To highlight the diversity of the datasets used for evaluating the benchmarks, we describe in the following table the mean and standard deviations for the size of the source datasets $N_R$, size of the target datasets $N_T$, number of features shared between the source and target datasets $D_s$, number of features private to the source datasets $D_{p_R}$, number of features private to the target datasets $D_{p_T}$, proportion of shared features in the source datasets $D_s / (D_s + D_{p_R})$, proportion of shared features in the target datasets $D_s / (D_s + D_{p_T})$ and the Maximum Mean Discrepancy (MMD) (computed using RBF kernel) between the shared features of the source and target datasets $\text{MMD}(X^{s_R}, X^{s_T})$. Note that the MMD between the shared features of the source and target datasets for MAGGIC is much higher than for TCGA and Twins, as the different datasets in MAGGIC represent medical studies with different patient populations.

---

> ### Author Response · Authors · 2022-08-02
> **Response to Reviewer 6zHZ [Part 1/3]**
>
> Thank you very much for your insightful comments and suggestions which have helped us to improve the paper!
>
> We provide below answers to the questions raised by the reviewer. Please note that the line references are for the revised manuscript. Moreover, please note that the changes in the revised manuscript are highlighted in blue.
>
> ### 1. Definition of transfer learning
>
> In our paper, we have followed the common definition of transfer learning [R1 - see Definition 1 (Transfer Learning)] where the conditional distribution of the outcome given the covariates changes between domains: $p(Y(w) \mid X^{R}) \neq p(Y(w) \mid X^{T})$. To be able to learn these changes in the outcome distribution in the transfer learning setting it is crucial to have labelled data in the target domain. Moreover, having heterogeneous feature spaces between the source and target domains further categorizes our setting as transfer learning.
>
> As opposed to us, [32] considers the setting where the source and target domains have the same feature space, the marginal distributions are different $p_R(X) \neq p_T(X)$, but the conditional label distribution stays the same $p_R(Y(w) \mid X) = p_T(Y(w) \mid X)$. This is why, using labelled data from the source domain and **unlabelled** data from the target domain, [32] aims to learn potential outcome functions that are invariant across the source and target domains. This setting is referred to as unsupervised domain adaptation, as also mentioned in [32].
>
> Nevertheless, we agree with the reviewer that our work has similarities with multi-task learning where the different domains represent different tasks and in fact, our model architecture is inspired by works in multi-task learning. However, these have been adapted to our heterogenous transfer setting for CATE estimation where we needed to handle the heterogenous feature space, transfer information between PO functions across domains and between PO functions within each source and target domain.
>
> In the revised manuscript, we have incorporated [R1] to clarify and justify our use of transfer learning for the problem we address in our paper.
>
> ### 2. Lines 229-230 (now 232-233)
> On the lines, we indeed mean + and not ||. This is because for layer L, we build the outputs $h\_{w, L}^{s}, h\_{w, L}^{p\_R}, h\_{w, L}^{p\_T}$ to each have the same dimension as the potential outcome $y$. Thus the potential outcomes for each domain are obtained by adding together the outcomes of the shared and private subspaces for continuous outcomes and by also applying a sigmoid function for binary outcomes (see lines 233-234). Please note that a similar approach is used by FlexTENet. We have now revised Section 4.2. to include this clarification about the dimensions of $h\_{w, L}^{s}, h\_{w, L}^{p\_R}, h\_{w, L}^{p\_T}$.
>
> ### 3. Figures 3.a and 3.b
>
> We have fixed the typos in Figure 3a and Figure 3b where we had the private encoders switched between the source and target domains. Moreover, on line 264, we have fixed the typo of having the treatment w concatenated with the private features of each domain. Thank you very much for pointing these out! Please let us know if there are other inconsistencies you were referring to.

---

### Official Review · Reviewer_ubFD · 2022-07-11

**Rating:** 6
**Confidence:** 5
**Soundness:** 3 good
**Presentation:** 3 good
**Contribution:** 2 fair

**Summary:**

This work aims to solve the heterogeneous transfer learning problem for CATE estimation by introducing several building blocks that use representation learning to handle the heterogeneous feature spaces and a flexible multi-task architecture with shared and private layers to transfer information between potential outcome functions across domains. Besides, they propose several building blocks to construct HTCE-learner, similar to the most common CATE learners.

**Questions:**

limited baseline models

**Limitations:**

To the best of my knowledge, the work is not as novel as the authors claim it is. Please refer to the following works:
* Pearl and Bareinboim, 2011, Transportability of Causal and Statistical Relations, https://ftp.cs.ucla.edu/pub/stat_ser/r372-a.pdf
* Bareinboim and Pearl, 2016, Causal inference and the data-fusion problem, https://ftp.cs.ucla.edu/pub/stat_ser/r450-reprint.pdf
* Magliacane et al, 2017, Causal Transfer Learning, https://www.semanticscholar.org/paper/Causal-Transfer-Learning-Magliacane-Ommen/b650e5d14213a4d467da7245b4ccb520a0da0312
* Mooij et al., 2016, Joint Causal Inference from Multiple Contexts, https://arxiv.org/abs/1611.10351

**Strengths And Weaknesses:**

Strengths:
1.  The paper is overall well written and it clearly defines the problem.
2.  These building blocks involve handling the heterogeneous feature spaces, sharing information between PO functions across domains and sharing information between PO functions within a single domain.

Weaknesses:
1. the model design is lack of innovation and major idea is based on the meta-learner.
2. limited baseline models.

---

> ### Author Response · Authors · 2022-08-02
> **Response to Reviewer ubFD [Part 3/3]**
>
> (cont’d)
>
> [R4] Magliacane et al, 2017, Causal Transfer Learning. Published with the following title (Domain Adaptation by Using Causal Inference to Predict Invariant Conditional Distributions)
>
> [R4] addresses the problem of domain adaptation for the **predictive** setting, where it considers labelled data in one or more source domains, **unlabelled** data in the target domain, same feature spaces between the source and target domains and aims to learn predictive functions that are **invariant** to the changes between domains to be able to reliably estimate the outcomes in the target domain. In particular, [R4] proposes tackling this unsupervised domain adaptation problem by modelling the different distributions between the source and target domains as different contexts of a single underlying system. The system variables are denoted as $X$ and the context variables are denoted as $C$. The context variables $C$ are used to model the interventions causing the distribution shifts between domains, such that the source and target domains come from different interventions on $C$. The aim is to predict the missing values of a target (outcome) variable in the target domain given the available source and target datasets. The paper solves this problem by finding a separating feature set A such that the label is conditionally independent of the context variables given $A$. This is because the distribution of $Y$ conditional on $A$ will then be invariant between the source and target domains and thus they can use a predictor from $A$ to $Y$ trained on the source domain to estimate $Y$ in the target domain.
>
> Our work is significantly different in the following ways: (1) we consider the problem of CATE estimation, (2) assume access to labelled data in both the source and target domains, (3) consider the case of heterogeneous feature space and (4) model different conditional distributions of the outcome given the features in the source and target domains. Our aim is to improve the estimation of CATE in the target domain by leveraging the shared structure with the source domains, while at the same time modelling the specific characteristic of the potential outcome functions in the target domain.
>
> [R5] Mooij et al., 2016, Joint Causal Inference from Multiple Contexts.
>
> Finally, [R5] proposes Joint Causal Inference (JCI), a new approach to **causal discovery** that uses multiple datasets from different contexts representing different types of interventions (the interventions in this setting represent perturbations to different variables in the dataset). In particular, [R5] aims to find the causal relationships among all variables in a system of interest. This is fundamentally different from our objective, as we are not aiming to perform causal discovery and build a causal graph, but rather to improve the estimation of CATE for a target domain of interest by leveraging data from a source domain.
>
>
> -------------------
>
> We thank the reviewer again for the insightful comments! Please let us know if you have any other concerns as we are eager to address them.

---

> > ### Comment · Reviewer_ubFD · 2022-08-08
> > **Thanks for your responses**
> >
> > My concerns have been solved. Thanks.

---

> ### Author Response · Authors · 2022-08-02
> **Response to Reviewer ubFD [Part 2/3]**
>
> ### Novelty and relationship to literature suggested by the reviewer
>
> Please allow us to clarify the novelty of our method and the relationship with the related works suggested by the reviewer. Firstly, please note that we acknowledge in the paper (Section 2) that the problem of transfer learning for treatment effect estimation has been previously considered. Our claim is to be the first work that addresses the problem of heterogeneous transfer, specifically for CATE estimation. This is different from the works suggested by the reviewer as described below.
>
> [R2] Pearl and Bareinboim, 2011, Transportability of Causal and Statistical Relations.
>
> [R3] Bareinboim and Pearl, 2016, Causal inference and the data-fusion problem.
>
> Both [R2] and [R3] aim to address the problem of the **identifiability** of causal effects for an observational dataset of interest by leveraging data from other datasets. In particular, [R2] investigates how to transfer **average** treatment (causal) effects obtained from experimental data (such as randomized clinical trials) to an observed population that may have different distribution of covariates, treatments and outcomes and where the causal relationship of interest cannot be identified using only the observational data. The paper assesses under which conditions such average causal effects can be transported according to the differences between the randomized and observational data. The authors also provide a brief discussion of how to transfer these average causal estimates between observational datasets. Alternatively, [R3] aims to help identify the **average** effects of interventions on a target population of interest by integrating multiple types of auxiliary data: data from a randomized study on the same population, data from an observational study on the same population, selection biased data from the same population and data from a randomized study from a different population. All of the auxiliary datasets have the same set of features as the target dataset. Both [R2] and [R3] consider this transportability problem of average treatment effects in the context of causal diagrams.
>
> This is different from our set-up where (1) we want to estimate **conditional average** treatment effects such that we can make personalized treatment recommendations based on the features of each patient. In addition, (2) we assume that the potential outcomes $Y(0)$ and $Y(1)$ **are identifiable** in both the source and target domains (see Assumptions 1 and 2). We also (3) handle the case of heterogeneous feature spaces and (4) only assume access to a source domain larger than a target domain, without putting any restrictions on whether this data is experimental or observational (see the experiment in Section 6.2 where we show that our method works well under different degrees of selection bias present in the source dataset).

---

> ### Author Response · Authors · 2022-08-02
> **Response to Reviewer ubFD [Part 1/3]**
>
> Thank you very much for your insightful comments and suggestions which have helped us to improve the paper!
>
> We provide below answers to the main concerns raised by the reviewer. Moreover, we have updated the discussion of related works (see Section 2 and in more details Appendix A) to incorporate the additional related methods provided by the reviewer. Please note that the changes in the revised manuscript are highlighted in blue.
>
> ### Model design innovation
>
> Please allow us to clarify what the contributions of our method are and where the novelty in our model design is coming from. To begin with, the main challenge we are aiming to address in this paper is to improve the estimation of conditional average treatment effects (CATE) for a target domain of interest by leveraging data from a source domain with a different feature space. To address this, we **do not** propose a single model, but rather introduce several building blocks that can be used to obtain heterogeneous transfer causal effect (HTCE-) learner equivalents for the most common and popular CATE-learners developed for binary treatments in a single patient population (such as the CATE-meta learners and TARNet based architectures). The motivation for this comes from the fact that the different CATE learners available have their own benefits and drawbacks in terms of the way they handle and the covariate shift induced by the selection bias in observational datasets and the inductive biases they use for modelling the PO functions within a single domain [R1]. Because of this, different CATE learners will result in better performance in different settings.
>
> Consequently, in this paper, our novelty comes from:
> * (1) proposing several building blocks (see Section 4) that can (a) handle the heterogeneous feature spaces, (b) share information between PO functions across domains $(\mu\_1^{R}$, $\mu\_1^{T})$ and $(\mu\_0^{R}$, $\mu\_0^{T})$ and (c) share information between PO function within each domain $(\mu\_0^{R}$, $\mu\_1^{R})$ and $(\mu\_0^{T}$, $\mu\_1^{T})$. Note that all of these are needed for heterogeneous transfer learning for CATE estimation.
> * (2) showing how these building blocks can be combined in a flexible way to obtain HTCE-equivalents of the common CATE-learners. We then demonstrate how our proposed HTCE-learners archive improved performance over the CATE-learners in a variety of scenarios relevant for transfer learning with heterogeneous feature spaces.
>
> [R1] Alicia Curth and Mihaela van der Schaar. Nonparametric estimation of heterogeneous treatment effects: From theory to learning algorithms. In International Conference on Artificial Intelligence and Statistics, pages 1810–1818. PMLR, 2021.

---

### Official Review · Reviewer_hxt3 · 2022-07-13

**Rating:** 7
**Confidence:** 2
**Soundness:** 3 good
**Presentation:** 3 good
**Contribution:** 4 excellent

**Summary:**

This paper addresses the problem of heterogeneous transfer learning for CATE estimation by using representation learning and a multi-task architecture to transfer information between potential outcome functions across domains, generalizing several existing CATE estimators to the transfer learning perspective.

**Questions:**

See weaknesses above.

**Limitations:**

Limitations and ethics are addressed appropriately. Some of the limitations and directions for future work discussed in the conclusion could also be mentioned more explicitly throughout the paper to make tradeoff decisions clear when introducing the framework.

**Strengths And Weaknesses:**

Strengths:
- The paper deals with an important problem that is understudied. I appreciate the motivation to study multitask architectures for CATE transfer learning when CATE has often been examined in isolated examples which real-world medical data provide compelling motivation for multitask and transfer learning settings.
- The paper effectively generalizes several CATE frameworks to the transfer learning setting which.
- The empirical results are sufficient and effectively show individual effects (e.g. the information sharing and the selection bias in Figure 6 and Figure 7), although not extensive experiments.

Weaknesses:
- The framework deals with fixed datasets with source and target domains. How does this framework extend to more complicated settings?  For example, what if we have multiple domains, the datasets are streaming, or the domains are unknown at training time? What about personalized feature spaces (e.g. different features are available for different samples)? Of course, I don’t expect all of these situations to be addressed in this manuscript but discussions about extensions would be helpful.

---

> ### Author Response · Authors · 2022-08-02
> **Response to Reviewer hxt3 [Part 2/2]**
>
> (cont'd)
>
> Another option could be the case of having full source datasets streaming, while the target dataset remains fixed. This would happen in the setting where for instance we gradually get access to data from multiple locations and we want to use these datasets as source datasets. In this scenario, one possibility would be to use the approach described above for having multiple source datasets and retraining a new model that incorporates all of the available source datasets as we get access to them. Another possibility would be to, instead of retraining a full model as we go from $M$ to $M+1$ source domains, we can add the needed private encoder and layers for the $(M+1)$-th domain and train only the new parameters and fine-tune the shared ones with the data from the new domain. While this is again outside our scope, it can provide interesting avenues for future work.
>
> ### Unknown domains at training time
>
> In this paper, we assume that the source and target domains are known. However, given that we handle the setting of transfer learning for heterogeneous feature spaces if the domains are unknown, one way to split available data into different domains, in this case, would be by using the same features to denote a single domain. For instance, if different patients in the dataset have recorded different sets of features, then the patients can be grouped according to having the same information collected for them and these can denote the different domains. Then, one needs to decide which represents the target domain, while keeping in mind that doing transfer learning is most beneficial when the target dataset is small (as highlighted by our experimental results in Section 6.2 and Figure 6 (bottom)).
>
> ### Personalized feature spaces
>
> In the case where different features are available for different patients and the source and target domains are unknown, then the patients can be grouped according to their feature spaces as described in the paragraph above. On the other hand, if the source and target datasets are pre-defined and the patients within each dataset have different features available for them, one possible option would be to consider the super-set of their features as the different feature spaces and consider the features that are not available for each patient as missing. However, as our HTCE-learners have not been designed for this particular setting of having missing features, one would also need to investigate if additional assumptions (in addition to ensuring that the no hidden confounders and overlap assumptions are still satisfied) are needed to be able to obtain valid estimates of causal effects.
>
> -------------------
>
> Overall, we believe that our paper handles an under-explored area for treatment effect estimation and could be a stepping stone in the development of more methods tailored to many of the scenarios described by the reviewer that have not yet been handled by methods for CATE estimation, despite their practical applicability.
>
> We thank the reviewer again for the insightful suggestions for future directions of our work. Please let us know if you were referring to different scenarios in your questions than the ones we discussed. Moreover, please let us know if you have any other concerns as we are eager to address them.

---

> > ### Comment · Reviewer_hxt3 · 2022-08-08
> > **Thank you for the responses.**
> >
> > Thank you for the thoughtful responses. After reading this response, along with the other reviews and author responses, I retain my recommendation of acceptance.
> >
> > Thank you for your detailed thoughts on the specific topics and extensions which I mentioned in the original review. While I find these specific topics interesting and meaningful as ways to think about extensions of the work, I don't intend to urge the authors to necessarily devote precious space in the manuscript to enumerate each particular topic. Rather, a broader discussion of the choices and tradeoffs made in this framework could be helpful. In this broader sense, I see similarities between my question and questions from other reviewers  about significance. Discussion of decisions, motivations, and extensions could improve the manuscript from both perspectives.

---

> ### Author Response · Authors · 2022-08-02
> **Response to Reviewer hxt3 [Part 1/2]**
>
> Thank you very much for your insightful comments and suggestions which have helped us to improve the paper!
>
> We agree with the reviewer that the manuscript would be significantly improved if we provide a more detailed discussion about potential extensions. We provide below detailed answers to the questions about potential extensions raised by the reviewer. Due to space limitations, it is not possible to add all of these to the revised paper. Thus, in our revised manuscript we have updated the Discussion section (Section 7) to briefly mention them, and we discuss them in more detail in the new Appendix F. Please note that the changes in the revised manuscript are highlighted in blue. We will incorporate more details about the extensions in the camera-ready version of the paper (which allows for an extra content page).
>
> ### Multiple domains
>
> While in this paper we mainly consider the standard and most common transfer learning setting of leveraging a source dataset to improve the estimation of outcomes on a target dataset of interest, our proposed approach could be easily extended to having multiple source domains, and in fact, would scale linearly to incorporating additional domains.
>
> More precisely, consider access to $M$ source domains, $\\{ D^{R_{m}}  \\}^{M}\_{m=1}$, where each source domain consists of $\\mathcal{D}^{R\_m} = \\{(X^{R\_m}\_i, W\_i, Y\_i)\\}\_{i=1}^{N\_{R\_{m}}}$ and a target domain as described in Section 3. The objective would still be the one from Section 3 of improving the estimation of CATE for the target domain, but in this case, we need to leverage data from all $M$ source domains. Our proposed building blocks could be extended as follows to handle the additional source domains. The building block for handling heterogeneous feature spaces between source and target domains could be extended by having encoders $\phi^{R\_m}$ for the private features $X^{p\_{R_m}}$ of each source domain (part of $X^{R\_m}$). Moreover, we can build an approach that shares information between the potential outcomes of each source domain with the target domain by having $L$ layers shared between all source domains and the target domains and $L$ private layers for each source domain. For source domain $m$, the input to the layer $(l+1)$-th can be obtained using $\tilde{h}\_{w, l+1}^{p\_{R\_m}} = [ h\_{w, l}^{s} || h\_{w, l}^{p\_{R\_m}}]$ where $h\_{w, l}^{p\_{R\_m}}$ is the output of the $l$-th private layer for source domain $m$ and $h\_{w, l}^{s}$ is the output of the $l$-th shared layer across all domains.
>
> While from a model development perspective this extension can be easily done, one also needs to consider whether the multiple source domains satisfy the underlying implicit assumptions for which such an architecture would be appropriate. In particular, it would be important to consider whether the PO functions across all source domains share information with the PO functions in the target domain, as using source domains that are significantly different from the target domain could harm performance.
>
> ### Datasets that are streaming
>
> We believe that there are multiple options for considering streaming datasets. One option could be the case where we already have a source domain and a target domain and we have incoming data streaming from the target domain. One such example in healthcare would be the case where in a single hospital we start collecting more or different clinical covariates for the patients which we now want to use for CATE estimation. In this case, the source domain would be the patients with only the original set of features, while the target domain would be the patients with the new set of features. As we start collecting these additional features, the initial target dataset will be small but it can start increasing with time as we observe more patients. In this setting, we can train an HTCE-learner with the initial data available from the source and target domains. However, instead of retraining the full HTCE-learner as we obtain more patients from the target domain, one option would be to fine-tune the weights using the incoming examples. While this is outside the scope of our paper, we believe that it would be important to investigate appropriate ways for performing such fine-tuning.

---

### Meta-Review · Area_Chair_kPKh · 2022-08-26

**Recommendation:** Accept
**Confidence:** Less certain

**Metareview:**

The paper studies methods for estimating conditional average treatment effects (CATE) under a shift in domain where source and target feature spaces are heterogenous. It is assumed that the (respective) CATEs in both source and target domains are identifiable through ignorability and overlap. No formal assumptions are made regarding the similarity of potential outcome distributions across domains, but implicitly that there exists a shared structure in the outcome functions. A number of heuristics are proposed to modify popular neural network CATE estimators to this setting, including a wide array of meta-learners such as propensity weighting, doubly robust estimators and TARNet.

Reviewers appreciated the setting of heterogenous feature domain adaptation which is understudied in the literature and representative of many transfer tasks of interest, such as transfer from a clinical trial to an observational cohort. Typically, the feature set collected in trials is smaller than in, say, a registry. However, as pointed out by one reviewer, the empirical evaluation does not consider such applications. In addition, no details are given in the main paper for how the heterogenous feature spaces are constructed for experiments (this is only given in the Appendix). The uniform sampling is quite unrealistic and most likely less challenging than real-world cases.

The authors make assumptions of ignorability and overlap, referring to previous work that this renders the causal effect identifiable. While this is true, the interesting complication in this work is that no assumptions are made regarding similarities of feature sets or outcome functions; these are left implicit. As a result, no claims can be made about the usefulness of source data for this task, see e.g., [1] for a discussion on hardness of transfer. In other words, the authors rely on empirical evidence to demonstrate this usefulness. In semi-synthetic experiments, the authors find that their proposed approach improves significantly over using only shared features, even when the number of target samples is minimal.

Reviewers were concern with the contextualisation of the work in the literature, given previous work on transportability of causal effects and on domain adaptation. Adding to this list, I would suggest that the authors refer to previous work on heterogenous-feature transfer learning. Under ignorability and overlap, the settings are not much different from each other, not least demonstrated by the fact that the T-learner solution performs well.

The authors propose several "building blocks" but don't evaluate the importance of these in isolation, using, for example, an ablation study. This makes it difficult to assess which components are necessary and which are not.

In summary, the considered setting is interesting and the algorithmic contributions appear useful empirically. The theoretical and methodological contributions are rather small, and the work should be better contextualised in the related topics of domain adaptation and transportability.

[1] Ben-David, Shai, and Ruth Urner. "On the hardness of domain adaptation and the utility of unlabeled target samples." International Conference on Algorithmic Learning Theory. Springer, Berlin, Heidelberg, 2012.

**Award:**

No

---

### Decision · Program_Chairs · 2022-09-14

Accept